# Improving Robustness of 3D Point Cloud Recognition from a Fourier Perspective

**Yibo Miao**[1,2], **Yinpeng Dong**[3,6†], **Jinlai Zhang**[4], **Lijia Yu**[5], **Xiao Yang**[3], **Xiao-Shan Gao**[1,2†]

[1] KLMM, Academy of Mathematics and Systems Science,
Chinese Academy of Sciences, Beijing 100190, China
[2] University of Chinese Academy of Sciences, Beijing 100049, China
[3] Tsinghua University, Beijing 100084, China
[4] Changsha University of Science and Technology, Changsha 410114, China
[5] Institute of Software, Chinese Academy of Sciences, Beijing 100190, China    [6] RealAI
miaoyibo@amss.ac.cn, dongyinpeng@tsinghua.edu.cn, xgao@mmrc.iss.ac.cn

## Abstract

Although 3D point cloud recognition has achieved substantial progress on standard benchmarks, the typical models are vulnerable to point cloud corruptions, leading to security threats in real-world applications. To improve the corruption robustness, various data augmentation methods have been studied, but they are mainly limited to the spatial domain. As the point cloud has low information density and significant spatial redundancy, it is challenging to analyze the effects of corruptions. In this paper, we focus on the frequency domain to observe the underlying structure of point clouds and their corruptions. Through graph Fourier transform (GFT), we observe a correlation between the corruption robustness of point cloud recognition models and their sensitivity to different frequency bands, which is measured by the GFT spectrum of the model's Jacobian matrix. To reduce the sensitivity and improve the corruption robustness, we propose Frequency Adversarial Training (FAT) that adopts frequency-domain adversarial examples as data augmentation to train robust point cloud recognition models against corruptions. Theoretically, we provide a guarantee of FAT on its out-of-distribution generalization performance. Empirically, we conducted extensive experiments with various network architectures to validate the effectiveness of FAT, which achieves the new state-of-the-art results.

## 1 Introduction

3D point cloud recognition based on deep neural networks (DNNs) [35, 36, 65] has achieved unprecedented performance on typical benchmarks [5, 67], which assume that the data are independently and identically distributed. However, in practical scenarios, point clouds suffer from severe corruptions (e.g., noise, density change, transformation) due to sensor imprecision and scene complexity [66, 76]. When the testing distribution is different from the training distribution caused by corruption, point cloud recognition models have significant performance degradation [41, 51], indicating that they lack the robustness of human visual system [40], while also raising concerns about safety and reliability of these models. As deep 3D point cloud recognition has been increasingly deployed in safety-critical applications, such as autonomous driving [6, 84], robotics [60, 92], and medical image processing [54], it is of crucial importance to improve the robustness of 3D point cloud recognition models to out-of-distribution (OOD) point cloud data induced by corruptions [10].

---

[†]Corresponding authors.

38th Conference on Neural Information Processing Systems (NeurIPS 2024).

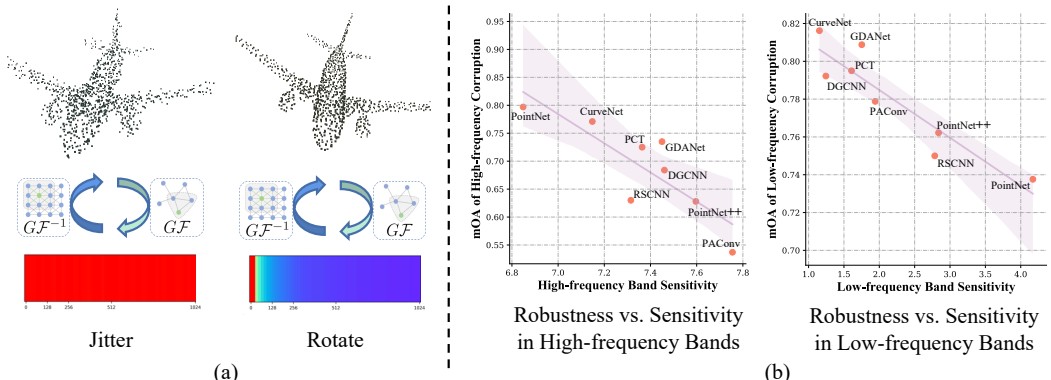

Figure 1: **(a):** The graph frequency-domain representations of "Jitter" and "Rotate" in ModelNet-C [41]. "Jitter" has higher power in the high-frequency region, while "Rotate" has higher power in the low-frequency region. **(b):** The relationship between the corruption robustness (measured by mean overall accuracy (mOA) [41]) of various models and the sensitivity to high/low frequency bands. Our proposed high/low frequency sensitivity metric is negatively correlated with the model's robustness under high/low frequency corruptions.

To improve the corruption robustness, the most effective approaches to date are based on carefully designed data augmentation techniques [41, 51]. Inspired by 2D image augmentations [85, 89], some methods blend two point clouds for data augmentation using shortest-path interpolation (e.g., Point-Mixup [7]), random blending (e.g., PointCutMix-R [90]), and rigid transformation (e.g., RSMix [24], PointCutMix-K [90]). PointWOLF [22] enriches data diversity by applying non-rigid deformation to object parts. WOLFMix [41] deforms objects first and then rigidly blends two deformed objects. Although these data augmentation techniques improve the corruption robustness to some extent, they are all based on spatial-domain transformations. Raw point clouds in the spatial domain have low information density and heavy spatial redundancy [8], making it challenging to analyze which specific information is corrupted. To address this challenge, we shift our attention from the spatial domain to the frequency domain to analyze the underlying structure of point clouds that is not easily observable from the raw point clouds. In the frequency domain, point clouds are compactly represented, facilitating a better understanding of low-level distortions that are free of high-level semantics.

To design robust models, the first step is to understand how corruption is represented in the frequency domain. We achieve this by transforming the raw point clouds and the corresponding corruptions into compact representations in the frequency domain using the graph Fourier transform (GFT) [43]. By visualizing the transformed signals, we observe that different corruptions affect varying frequency bands, as shown in Fig. 1(a). Motivated by the differences, we investigate the relationship between the corruption robustness of various point cloud recognition models and their sensitivity to different frequency bands [81]. To measure the sensitivity, we design **a novel metric based on GFT spectrum of the Jacobian matrix** of the model, as shown in Fig. 3. Our key insight is that **our proposed high/low frequency sensitivity metric is negatively correlated with the model's robustness under high/low frequency corruptions**, as shown in Fig. 1(b). This correlation emphasizes the importance of the model's sensitivity to high and low frequencies for corruption robustness. However, it is still challenging to simultaneously reduce the sensitivity of point cloud recognition models to both high and low frequencies.

To address this issue, we propose **Frequency Adversarial Training (FAT)** to improve the corruption robustness of 3D point cloud recognition models. FAT trains a model with adversarial examples that add perturbations to the frequency-domain representations of point clouds. Intuitively, a model robust to worst-case perturbations should be more resistant to real-world corruptions [72, 80]. We provide **a theoretical analysis that demonstrates the effectiveness of FAT in ensuring OOD generalization of the model**, as shown in Theorem 1. To eliminate potential performance degradation due to mutual interference between high and low frequency signals, we utilize the AdvProp training framework [72], based on which we use three separate batch normalization (BN) statistics for clean samples, high-frequency adversarial samples, and low-frequency adversarial samples, respectively.

We conducted extensive experiments to validate the effectiveness of our approach in improving the robustness of point cloud recognition models under common corruptions [41, 51]. With various

network architectures, our method improves the corruption robustness by a large margin. By integrating our approach with previous data augmentation techniques, we achieve the new state-of-the-art performance.

## 2   Related work

**Deep learning on 3D point clouds.** Deep 3D point cloud recognition [16, 35, 38, 55, 70, 73, 79] has emerged in recent years as a prominent research area with diverse applications in several fields such as 3D object classification [46, 83, 86], 3D scene segmentation [20, 64, 75], and 3D object detection in autonomous driving [77, 95]. One of the pioneering works is PointNet [35], which employs a multilayer perceptron to learn point features and utilizes a max-pool module to aggregate them efficiently. Many subsequent works [13, 30, 36, 78] improve upon PointNet. Several approaches focus on designing special convolutions on 3D domains [26, 31, 56] or developing graph neural networks [14, 44, 65] to improve point cloud recognition, such as DGCNN [65] which builds a dynamic graph for point cloud data. Recently, drawing inspiration from research in the frequency domain [4, 49, 61, 81], GDANet [74] introduces a geometry-disentangle module to dynamically separate point clouds into the contour and flat parts of 3D objects, thereby capturing complementary 3D geometric semantics. PCT [17] uses Transformer to improve point cloud learning. Additionally, there is a growing discussion on point cloud augmentation, including mix-based augmentations [7, 90], deformation-based augmentations [22], and auto-augmentations [25].

**Robustness in 3D point cloud recognition.** Following the previous studies on robustness in the 2D image domain [53, 3, 15, 19, 32, 42, 59, 9, 82], several works [18, 50, 52, 63, 69, 34, 97] have explored the robustness of 3D point cloud classifiers. Concerning adversarial robustness, Xiang et al. [69] first demonstrate that point cloud recognition is vulnerable to adversarial point generation attacks. Further research [21, 28, 29, 57, 91, 2] has employed gradient-based point perturbation attacks. Some defensive techniques are proposed, such as input randomization [12, 93] and geometry-aware framework [68] to defend against such vulnerabilities. Sun et al. [47, 48] have studied the effectiveness of adversarial training and pre-training on self-supervised tasks in enhancing robustness. In terms of corruption robustness, some works have studied the problem using invariant feature extraction [71], and adaptive sampling [76]. Recently, two benchmarks [41, 51] are developed for the robustness of 3D point cloud recognition under corruptions and demonstrate the effectiveness of data augmentation. However, unlike the existing spatial-domain data augmentation techniques [22, 25], in this paper, we focus on the frequency domain and propose Frequency Adversarial Training (FAT) to improve the model's out-of-distribution generalization ability.

## 3   Methodology

The existing 3D point cloud recognition models exhibit significant performance degradation under point cloud corruptions [41, 51]. Although data augmentation techniques have shown the effectiveness in improving robustness, they are typically based on spatial-domain transformations, which suffer from low information density and heavy spatial redundancy of the raw point clouds. Consequently, it is difficult to analyze which specific information has been lost due to corruptions within the spatial domain. To address this challenge, we shift our focus to the frequency domain, which enables us to analyze the underlying structure of point clouds.

In the following, we first provide the background knowledge of graph Fourier transform (GFT) in Sec. 3.1, then analyze the point cloud corruptions in the frequency domain in Sec. 3.2, and investigate the relationship between the model's corruption robustness and sensitivity to frequency changes in Sec. 3.3. Based on the analyses, we propose a Frequency Adversarial Training (FAT) method detailed in Sec. 3.4 with a theoretical analysis to guarantee its effectiveness in Sec. 3.5.

### 3.1   Graph Fourier transform

Images are typically transformed and recovered in the frequency domain with the 2D discrete Fourier transform (DFT) and inverse DFT [39]. Unlike images, although 3D point clouds are highly structured, they reside on irregular domains without an ordering of points, hindering the deployment of traditional Fourier transforms. However, graphs provide a natural and accurate representation of

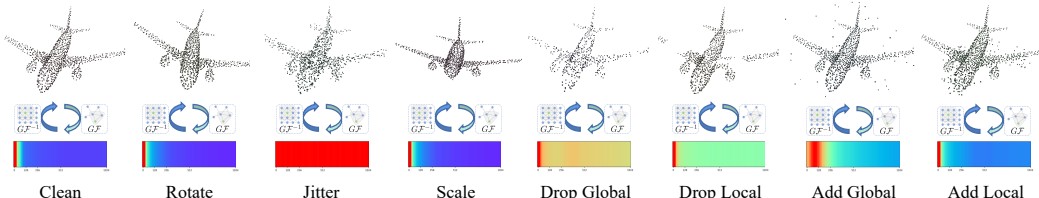

Figure 2: The leftmost image displays the graph frequency-domain representation of the raw point clouds. To estimate the expected value of $\mathbb{E}_{\mathcal{P}}[\|G\mathcal{F}(\mathcal{P})\|]$, we average over all validation point clouds in ModelNet40 [67]. The frequencies are arranged from left to right in ascending order. The other seven images display the graph frequency-domain representations of each corruption in ModelNet-C [41]. The raw point clouds exhibit higher power in the low-frequency region. The corruption "Jitter" has much higher power in the high-frequency region. The power of corruptions such as "Rotate" and "Scale" is concentrated on the low-frequency components.

irregular point clouds. Once a graph is constructed to represent the point cloud, the graph Fourier transform (GFT) [43] can compactly transform it into the frequency domain.

Given a point cloud $\mathcal{P} := \{\boldsymbol{p}_i\}_{i=1}^n \in \mathbb{R}^{n \times 3}$ of $n$ points, where $\boldsymbol{p}_i$ denotes the xyz coordinates of a point, we construct a directed graph $\mathcal{G} = \{\mathcal{P}, \mathcal{E}, \boldsymbol{W}\}$ to represent it. The graph consists of a vertex set $\mathcal{P}$, an edge set $\mathcal{E}$ connecting the vertices, and an adjacency matrix $\boldsymbol{W}$. The entry $w_{i,j}$ in the adjacency matrix represents the weight of the edge from vertices $i$ to $j$, which is used to capture the similarity between adjacent vertices. Here, we construct a weighted $k$-nearest neighbor graph (i.e., each vertex is only connected to its $k$-nearest neighbors) using the Euclidean distance $d_{ij} = \|\boldsymbol{p}_i - \boldsymbol{p}_j\|_2$ between vertices $i$ and $j$, and the weight of the edge is $w_{i,j} = e^{-d_{ij}^2}$.

After constructing the graph representation of the point cloud, we focus on the combinatorial graph Laplacian [45], defined as $\boldsymbol{L} := \boldsymbol{D} - \boldsymbol{W}$, where $\boldsymbol{D}$ is a diagonal matrix with the $i$-th diagonal entry $d_{i,i} = \sum_{j=1}^n w_{i,j}$ representing the degree of the $i$-th node. $\boldsymbol{L}$ is symmetric and positive semi-definite, and can be eigen-decomposed as $\boldsymbol{L} = \boldsymbol{U}\boldsymbol{\Lambda}\boldsymbol{U}^\top$, where $\boldsymbol{U} = [\boldsymbol{u}_1, ..., \boldsymbol{u}_n]$ is an orthogonal matrix containing the eigenvectors $\boldsymbol{u}_i$, and $\boldsymbol{\Lambda} = \mathrm{diag}(\lambda_1, ..., \lambda_n)$ is a diagonal matrix containing the eigenvalues. The eigenvalues are sorted in ascending order, representing frequencies from low to high. For a point cloud $\mathcal{P}$, the graph Fourier transform (GFT) can be applied to transform it into a compact representation in the frequency domain: $\hat{\mathcal{P}} = G\mathcal{F}(\mathcal{P}) := \boldsymbol{U}^\top\mathcal{P}$. The low-frequency components represent the coarse shape of the point cloud, while the high-frequency components represent the fine details. The inverse graph Fourier transform (IGFT) can be used to recover the point cloud in the spatial domain as $\mathcal{P} = G\mathcal{F}^{-1}(\hat{\mathcal{P}}) := \boldsymbol{U}\hat{\mathcal{P}}$.

## 3.2 Analyzing point cloud corruptions in the frequency domain

We employ GFT to transform point clouds and their corruptions into compact representations in the frequency domain, allowing us to analyze the underlying structures of these low-level distortions that are hardly observable in the spatial domain. For raw point clouds, we transform them to the frequency-domain representations and calculate $\mathbb{E}_{\mathcal{P}}[\|G\mathcal{F}(\mathcal{P})\|]$ by averaging over all validation point clouds in ModelNet40 [67]. For each corruption type in ModelNet-C [41], we calculate $\mathbb{E}_{\mathcal{P}}[\|G\mathcal{F}(\mathcal{C}(\mathcal{P}) - \mathcal{P})\|]$ similarly, where $\mathcal{C}$ denotes the corruption function. As the input point clouds have three spatial axes $(x, y, z)$, we take the average over these channels. In Fig. 2, we visualize the graph frequency-domain representations of raw point clouds and the corruptions in ModelNet-C. We can see that the raw point clouds have higher power in the low-frequency region, while the corruption "Jitter" leads to higher power in the high-frequency region. For corruptions such as "Rotate" and "Scale", the corrupted power is concentrated more on the low-frequency components. The results demonstrate that different corruptions of point clouds affect different frequency bands.

## 3.3 Relationship between corruption robustness and sensitivity to frequency bands

Motivated by the different effects of corruptions on varying frequency bands observed in the graph frequency-domain representations, we investigate the relationship between the corruption robustness of 3D point cloud recognition models and their sensitivity to different frequency bands.

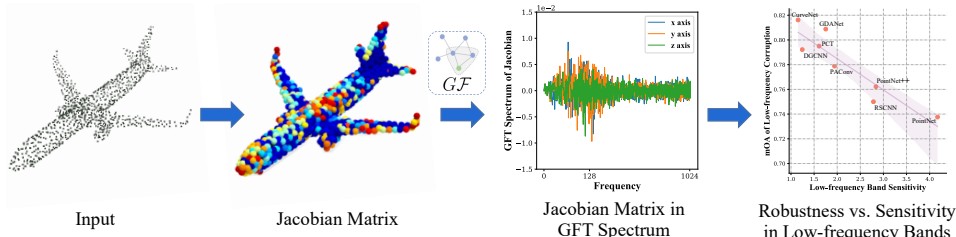

| Input | Jacobian Matrix | Jacobian Matrix in GFT Spectrum | Robustness vs. Sensitivity in Low-frequency Bands |

Figure 3: An illustration of computing the Fourier spectrum of the Jacobian matrix for a single input point cloud. First, the Jacobian matrix for the input point cloud is computed. The gradient value of the output loss is visualized for each point. A higher gradient value (skewed to red) indicates that the model is more sensitive to changes at that point. Next, we utilize Graph Fourier Transform (GFT) on the Jacobian matrix to obtain a compact representation and measure its sensitivity in the Fourier domain. Finally, by examining the sensitivity measurement of different point cloud models in different frequency bands, we construct a relationship diagram with natural robustness.

To measure the sensitivity of a model on different frequency bands, we propose to perform graph Fourier transform (GFT) on the Jacobian matrix of the model's output loss with respect to its input point cloud. Intuitively, the Jacobian matrix represents how the model's output changes with small variations in its input point cloud, revealing its sensitivity to different points in the spatial domain [1]. With GFT, we can obtain the frequency-domain representation of the Jacobian matrix, which reveals the model's sensitivity to different frequency bands of input. If a model's Jacobian matrix has a high proportion of low/high frequency components, it will be sensitive to low/high frequency bands.

Fig. 3 illustrates the computation of the frequency-domain Jacobian matrix for a single point cloud. Specifically, given an input point cloud $\mathcal{P}$, a classification model $h$, and a standard cross-entropy loss function $\mathcal{L}_h$ for the classification task, the Jacobian matrix $\mathcal{J}(\mathcal{P}) := \nabla_{\mathcal{P}} \mathcal{L}_h$ of the loss with respect to the input point cloud can be calculated. We then perform GFT on $\mathcal{J}(\mathcal{P})$ to obtain its frequency-domain representation, denoted as $\widehat{\mathcal{J}(\mathcal{P})} = \boldsymbol{U}^\top \mathcal{J}(\mathcal{P})$ in a compact form, using the original point cloud's neighborhood relations and feature vector matrix. Since the input point cloud has three axis channels $(x, y, z)$, we take the average of these channels and normalize the result. We measure the model's sensitivity to input perturbations in the low-frequency band by summing the squares of the amplitudes of the first $\lambda$ frequencies of the Jacobian matrix's graph Fourier spectrum. The sensitivity to high-frequency perturbations is measured by summing the squares of the amplitudes of the remaining $1024 - \lambda$ frequencies. A higher value of the metric indicates greater sensitivity to perturbations in that frequency band.

We can now measure the importance of the sensitivity to different frequency bands of point cloud recognition models on their corruption robustness. First, we measure and establish the relationship between sensitivity to high/low frequency bands of different point cloud models and their accuracy under high/low frequency corruptions. As illustrated in Fig. 1(b), our proposed frequency sensitivity metrics are negatively correlated with the corruption robustness. Therefore, point cloud models that are less sensitive to high/low frequency bands exhibit better robustness to high/low frequency corruptions. This correlation indicates that the sensitivity of models to different frequency bands affects their corruption robustness, providing insights for further improving the robustness of point cloud recognition models.

### 3.4 Frequency adversarial training

The above analyses highlight the importance of the sensitivity of point cloud recognition models to high and low frequencies on their corruption robustness. However, reducing the sensitivity of point cloud models to both high and low frequencies is still challenging. To address this problem, we propose **Frequency Adversarial Training (FAT)** to improve the corruption robustness of point cloud recognition models using adversarial examples in the frequency domain. Intuitively, a model trained to be robust to worst-case adversarial perturbations should be naturally robust to real-world corruptions [72, 80], as also theoretically demonstrated in Sec. 3.5.

To simultaneously reduce the sensitivity of point cloud recognition models to high and low frequencies, we generate high-frequency adversarial examples and low-frequency adversarial examples, which are added to the training set. We generate high-frequency adversarial examples that alter the details of

the point clouds, and low-frequency adversarial examples that change the rough shapes of the point clouds. To prevent the mutual interference of high-frequency and low-frequency adversarial examples that may lead to a decrease in model performance, we adopt the AdvProp training framework [72], where clean samples, high-frequency adversarial samples, and low-frequency adversarial samples are separately processed using three batch normalizations during adversarial training. Specifically, for an input point cloud $\mathcal{P}$ with the ground-truth label $y$, our optimization objective is

$$
\begin{aligned}
\arg \min_{\theta} \Big[ \mathbb{E}_{(\mathcal{P}, y) \sim \mathbb{D}} \Big( &\mathcal{L}_h(\theta, \mathcal{P}, y) + \max_{\epsilon_h \in \mathbb{S}_h} \mathcal{L}_h(\theta, G\mathcal{F}^{-1}(G\mathcal{F}(\mathcal{P}) + \epsilon_h), y) \\
&+ \max_{\epsilon_l \in \mathbb{S}_l} \mathcal{L}_h(\theta, G\mathcal{F}^{-1}(G\mathcal{F}(\mathcal{P}) + \epsilon_l), y) \Big) \Big],
\end{aligned}
\tag{1}
$$

where $\mathbb{D}$ is the underlying data distribution, $\mathcal{L}_h$ is the loss function, $\theta$ is the network parameter, $\epsilon_h$ and $\epsilon_l$ are high-frequency and low-frequency adversarial perturbations, and $\mathbb{S}_h$ and $\mathbb{S}_l$ are the high-frequency and low-frequency perturbation ranges, respectively.

### 3.5 Theoretical analysis

To verify the claim that a model robust to frequency-domain worst-case perturbations should be more resistant to real-world corruptions, we provide a theoretical analysis that demonstrates the effectiveness of FAT in ensuring OOD generalization of the model.

Suppose $(x, y)$ is a pair of training sample $x$ and its label $y$. The loss on $(x, y)$ with model parameter $\theta$ is $\mathcal{L}(\theta, (x, y))$, where $\mathcal{L}(\theta, (x, y))$ is continuous and differentiable for both $\theta$ and $(x, y)$. We let $f(x) := \mathcal{L}(\theta, (x, y))$ for simplicity. Let $\mathcal{F}$ and $\mathcal{F}^{-1}$ denote the Fourier transform and inverse Fourier transform, respectively. The norm $\| \cdot \|_p$ denotes the $\ell_p$-norm. We have the following theorem:

**Theorem 1.** *If $f$ satisfies that: $f(x) \in [0, M]$ for all $x$, $|f(\mathcal{F}^{-1}(\mathcal{F}(x) + \alpha)) - f(x)| \leq \epsilon$ for all $x$ and $\|\alpha\|_p \leq \delta$, then for any distribution $P_o$ and $P_a$ satisfying that $Was^p(P_o, P_a) := (\inf_{u \in \Pi(P_o, P_a)} \mathbb{E}_{(x,z) \sim u}[\|\mathcal{F}(x) - \mathcal{F}(z)\|_p^p])^{1/p} \leq \eta$, where $\eta < \delta$, then, with probability $1 - \gamma$, we have:*

$$
\mathbb{E}_{z \sim P_o}[f(z)] - \frac{1}{m} \sum_{i=1}^{m} f(x_i) \leq \epsilon \left( 1 - \frac{\eta^p}{\delta^p} - \sqrt{\frac{\ln(4/\gamma)}{2m}} \right) + \frac{\eta^p}{\delta^p} M + 4M \sqrt{\frac{\ln(4/\gamma)}{2m}}, \tag{2}
$$

*where $\{x_i\}_{i=1}^{m}$ are i.i.d. samples from $P_a$.*

**Remark 1.** *Intuitively, OOD corresponds to the shifted distribution $P_o$ that approaches the training distribution $P_a$. Thus $Was^p(P_o, P_a)$ defines OOD from the perspective of measuring the distance between distributions. $\mathbb{E}_{z \sim P_o}[f(z)] - \frac{1}{m} \sum_{i=1}^{m} f(x_i)$ represents the OOD generalization error of the model. $|f(\mathcal{F}^{-1}(\mathcal{F}(x) + \alpha)) - f(x)| \leq \epsilon$ and $\|\alpha\|_p \leq \delta$ indicate that the model is robust under frequency-domain perturbations. The bound (2) implies that models that are adversarially robust in the frequency-domain have smaller generalization bounds on OOD data.*

The proof of Theorem 1 is deferred to Appendix A. Thus, the frequency-domain adversarial robustness of the model guarantees the generalization on OOD data. We have the following observations:

- The right-hand side of Eq. (2) implies that models that are more robust to frequency domain adversarial samples (i.e., larger $\delta$ and smaller $\epsilon$) have smaller OOD generalization bounds and thus perform better on OOD data.

- For Eq. (2), a larger number of training samples $m$ leads to a smaller OOD generalization bound. This indicates that more training samples can compensate for the degradation of generalization performance.

## 4 Experiments

In this section, we first detail the experimental settings in Sec. 4.1, then present the main results in Sec. 4.2 to show the effectiveness of our method. We further integrate our method with other data augmentation techniques in Sec. 4.3 and perform ablation studies in Sec. 4.4.

Table 1: Quantitative results of vanilla training, adversarial training, DUP Defense and our proposed Frequency Adversarial Training (FAT) on the ModelNet-C test set. Our proposed FAT outperforms all other methods in terms of mean corruption error (mCE), which demonstrates the effectiveness of FAT for improving corruption robustness.

|  | Method | OA ↑ | mCE ↓ | Rotate | Jitter | Scale | Drop-G | Drop-L | Add-G | Add-L |
|---|---|---|---|---|---|---|---|---|---|---|
| DGCNN | Vanilla Training | 0.926 | 1.000 | 1.000 | 1.000 | 1.000 | 1.000 | 1.000 | 1.000 | 1.000 |
|  | Adv Training | 0.925 | 0.926 | 1.019 | 0.582 | 1.043 | 0.996 | 1.101 | 0.871 | 0.869 |
|  | DUP Defense | 0.906 | 0.905 | 1.112 | 0.902 | 1.181 | 1.048 | 1.483 | 0.285 | 0.327 |
|  | FAT (Ours) | 0.925 | **0.825** | 0.898 | 0.453 | 0.989 | 0.931 | 0.971 | 0.773 | 0.760 |
| PointNet | Vanilla Training | 0.907 | 1.422 | 1.902 | 0.642 | 1.266 | 0.500 | 1.072 | 2.980 | 1.593 |
|  | Adv Training | 0.904 | 1.372 | 1.851 | 0.563 | 1.287 | 0.448 | 1.077 | 2.888 | 1.487 |
|  | DUP Defense | 0.876 | 1.246 | 2.088 | 0.668 | 1.649 | 0.802 | 1.396 | 1.649 | 1.153 |
|  | FAT (Ours) | 0.902 | **1.237** | 1.553 | 0.370 | 1.606 | 0.448 | 1.097 | 2.583 | 1.004 |
| PCT | Vanilla Training | 0.930 | 0.925 | 1.042 | 0.870 | 0.872 | 0.528 | 1.000 | 0.780 | 1.385 |
|  | Adv Training | 0.919 | 0.976 | 1.042 | 0.389 | 1.074 | 0.911 | 1.193 | 1.108 | 1.116 |
|  | DUP Defense | 0.919 | 0.925 | 1.112 | 0.699 | 1.043 | 0.738 | 1.261 | 0.410 | 1.215 |
|  | FAT (Ours) | 0.920 | **0.907** | 1.009 | 0.345 | 1.085 | 0.843 | 1.237 | 0.912 | 0.920 |
| GDANet | Vanilla Training | 0.934 | 0.892 | 0.981 | 0.839 | 0.830 | 0.794 | 0.894 | 0.871 | 1.036 |
|  | Adv Training | 0.926 | 0.960 | 1.112 | 0.506 | 1.032 | 0.927 | 1.140 | 1.064 | 0.938 |
|  | DUP Defense | 0.915 | 0.897 | 1.140 | 0.788 | 1.064 | 0.698 | 1.179 | 0.427 | 0.985 |
|  | FAT (Ours) | 0.928 | **0.850** | 1.167 | 0.408 | 0.926 | 0.794 | 1.111 | 0.654 | 0.887 |

## 4.1 Experimental setup

**Dataset.** To validate the effectiveness of our FAT method in enhancing the corruption robustness of 3D point cloud recognition models, we train all models on the standard ModelNet40 training set [67]. In addition to reporting the performance of the models on the original ModelNet40 validation set, we also evaluate the corruption robustness on ModelNet-C [41] in the main paper and ModelNet40-C [51] in Appendix B. The ModelNet40 dataset [67] contains 12,311 CAD models with 40 common object categories in the real world. We use the official split [35], where 9,843 examples are used for training and the remaining 2,468 examples are used for testing. The ModelNet-C dataset [41] is designed for measuring the network robustness to common point cloud corruptions. It consists of 7 different corruption types, including "Scale", "Jitter", "Rotate", "Drop Global", "Drop Local", "Add Global", and "Add Local". Each type of corruption has five severity levels. ModelNet40-C [51] is a similar dataset with 15 corruptions, which will be detailed in Appendix B.

**Model architectures.** Following [41, 51], we select four representative model architectures: PointNet [35], DGCNN [65], PCT [17], and GDANet [74]. These models represent different architectural designs and have been widely applied in 3D visual tasks.

**Evaluation metrics.** To measure the corruption robustness of different methods, we follow [41] and use the mean corruption error (mCE) as the main evaluation metric. We adopt the official baseline DGCNN and first compute the corruption error (CE) for a given corruption type $i$ by averaging over 5 severity levels: $\text{CE}_i = \frac{\sum_{l=1}^{5}(1-\text{OA}_{i,l})}{\sum_{l=1}^{5}(1-\text{OA}_{i,l}^{\text{DGCNN}})}$, where $\text{OA}_{i,l}$ is the overall accuracy on a corruption test set $i$ at severity level $l$, and $\text{OA}_{i,l}^{\text{DGCNN}}$ is the overall accuracy of the baseline. Then, we average over the 7 corruption types to compute the mean corruption error: $\text{mCE} = \frac{1}{N}\sum_{i=1}^{N}\text{CE}_i$. In addition, we also report the clean overall accuracy (OA), the corruption overall accuracy (mOA), and the relative mCE (RmCE) following [41]. Due to space constraints, we provide the definition of RmCE and report mOA and RmCE in Appendix B.

**Implementation details.** For each method, we train 250 epochs using the smooth cross-entropy loss [65] and Adam optimizer [23], and select the best performant model for further evaluation. We follow the DGCNN protocol [16]. For our method, we set $k = 30$ for the $k$-nearest neighbor graph and $\lambda = 100$ for dividing high-frequency and low-frequency [29]. We use PGD [33] and AOF [27] to generate high-frequency adversarial examples and low-frequency adversarial examples, respectively. We constrain $\mathbb{S}_h$ and $\mathbb{S}_l$ by 0.3 and 0.5, respectively. For more detailed training settings, please refer to Appendix B.

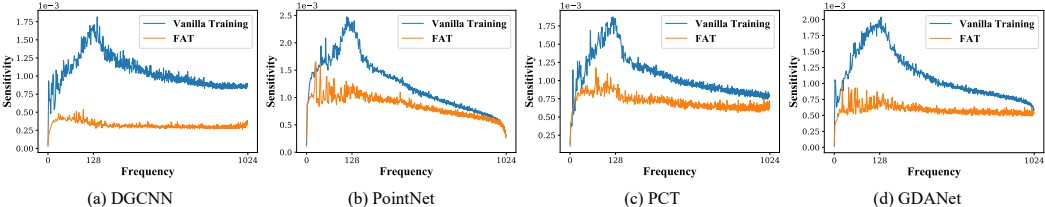

|     |     |     |     |
| --- | --- | --- | --- |
| (a) DGCNN | (b) PointNet | (c) PCT | (d) GDANet |

Figure 4: Visualization of the sensitivity maps based on Jacobian matrices of Frequency Adversarial Training (FAT) and vanilla training under four different model architectures. FAT reduces the model sensitivity to different frequency bands, thereby enhancing their robustness to corruptions.

## 4.2 Main results

In this section, following [41, 51], we compare our proposed Frequency Adversarial Training (FAT) method with vanilla training, adversarial training and DUP Defense [93] on the ModelNet-C test set, demonstrating the effectiveness of FAT in enhancing corruption robustness. Table 1 presents a comparative analysis of different methods based on mean corruption error (mCE), clean overall accuracy (OA), and corruption error (CE) for each corruption type.

As shown in Table 1, our proposed Frequency Adversarial Training (FAT) outperforms all other methods in terms of mean corruption error (mCE), while exhibiting comparable performance in terms of overall accuracy (OA). The improvement in corruption robustness across the four different model architectures demonstrates the generalizability/universality of our method across different architectures. In Fig. 4, we visualize the sensitivity maps based on Jacobian matrices of Frequency Adversarial Training (FAT) and vanilla training under four different model architectures. FAT reduces the sensitivity of the model across different frequency bands.

It is noteworthy that GDANet introduces a geometry-disentangle module to dynamically disentangle point clouds into the contour and flat part of 3D objects, capturing complementary 3D geometric semantics. In contrast, FAT does not modify the network architecture to focus on the frequency domain but instead employs adversarial training in the frequency domain. As shown in Table 1, the two methods are complementary and synergistic, leading to improved model robustness. We report the performance of different methods in terms of overall corruption accuracy (mOA) and relative mCE (RmCE) in Appendix B, where the improvement in robustness of FAT is also significant under these metrics. The comparisons in Table 1 and Appendix B confirm that our proposed FAT enhances the OOD generalization ability of the model.

## 4.3 Data augmentation

To further validate the effectiveness of our proposed Frequency Adversarial Training (FAT), following [41], we investigate the performance of FAT in combination with different data augmentation strategies, including RSMix [24], PointWOLF [22], and WOLFMix [41]. These strategies respectively represent mix-based augmentation, deformation-based augmentation, and a combination of both mix-based and deformation-based augmentation. RSMix involves rigidly blending two point clouds using a transformation. PointWOLF enriches data diversity by applying non-rigid deformations to object parts. WOLFMix, designed based on PointWOLF and RSMix, first deforms the objects and then rigidly blends two deformed objects. When combining the data augmentation strategies, we first perform data augmentation on the input point cloud and then generate adversarial examples. For mix-based augmentation, we perform untargeted adversarial attacks on both labels being mixed to generate the adversarial examples.

In Table 2, we show the performance of FAT when integrated with different data augmentation strategies in terms of mean corruption error (mCE), clean overall accuracy (OA), and corruption error (CE) for each corruption type. Compared with a single data augmentation strategy, the combination of FAT and data augmentation strategies achieves a better mCE, which is attributed to the complementary and compatible information from both the spatial and frequency domains. The improvement in corruption robustness under three different data augmentation strategies demonstrates the generalization capability of our proposed method. As shown in Table 2, training GDANet with the combination of our proposed FAT with WOLFMix achieves a new state-of-the-art performance, with an impressive 0.537 mCE.

Table 2: Quantitative results of combining FAT with different data augmentation strategies on the ModelNet-C test set. Compared with a single data augmentation strategy, the combination of FAT and different data augmentation strategies achieves a better mCE. Training GDANet with the combination of our proposed FAT with WOLFMix achieves the new state-of-the-art performance, with an impressive **0.537 mCE**.

| | Method | OA ↑ | mCE ↓ | Rotate | Jitter | Scale | Drop-G | Drop-L | Add-G | Add-L |
|---|---|---|---|---|---|---|---|---|---|---|
| PointNet | Vanilla Training | 0.907 | 1.422 | 1.902 | 0.642 | 1.266 | 0.500 | 1.072 | 2.980 | 1.593 |
| | RSMix | 0.902 | 1.276 | 1.372 | 0.532 | 2.234 | 0.593 | 1.145 | 2.241 | 0.815 |
| | RSMix+FAT (Ours) | 0.904 | **1.084** | 1.340 | 0.389 | 1.670 | 0.415 | 0.899 | 2.241 | 0.636 |
| | PointWOLF | 0.902 | 1.311 | 0.912 | 0.633 | 2.128 | 0.754 | 1.575 | 2.210 | 0.964 |
| | PointWOLF+FAT (Ours) | 0.902 | **0.993** | 0.558 | 0.487 | 1.372 | 0.589 | 1.411 | 1.759 | 0.775 |
| | WOLFMix | 0.880 | 1.149 | 0.986 | 0.560 | 2.096 | 0.605 | 1.155 | 1.854 | 0.789 |
| | WOLFMix+FAT (Ours) | 0.882 | **0.945** | 0.726 | 0.491 | 1.691 | 0.520 | 1.048 | 1.498 | 0.644 |
| PCT | Vanilla Training | 0.930 | 0.925 | 1.042 | 0.870 | 0.872 | 0.528 | 1.000 | 0.780 | 1.385 |
| | RSMix | 0.925 | 0.660 | 1.116 | 0.614 | 1.106 | 0.488 | 0.522 | 0.302 | 0.473 |
| | RSMix+FAT (Ours) | 0.925 | **0.604** | 1.093 | 0.354 | 1.106 | 0.427 | 0.531 | 0.308 | 0.411 |
| | PointWOLF | 0.923 | 0.846 | 0.428 | 0.788 | 0.979 | 0.504 | 1.130 | 1.040 | 1.051 |
| | PointWOLF+FAT (Ours) | 0.923 | **0.785** | 0.465 | 0.415 | 1.096 | 0.556 | 1.217 | 0.953 | 0.796 |
| | WOLFMix | 0.922 | 0.585 | 0.442 | 0.788 | 0.989 | 0.444 | 0.546 | 0.319 | 0.564 |
| | WOLFMix+FAT (Ours) | 0.920 | **0.570** | 0.572 | 0.326 | 1.351 | 0.444 | 0.560 | 0.325 | 0.415 |
| GDANet | Vanilla Training | 0.934 | 0.892 | 0.981 | 0.839 | 0.830 | 0.794 | 0.894 | 0.871 | 1.036 |
| | RSMix | 0.927 | 0.680 | 1.205 | 0.873 | 1.000 | 0.484 | 0.531 | 0.281 | 0.385 |
| | RSMix+FAT (Ours) | 0.929 | **0.617** | 1.153 | 0.427 | 1.021 | 0.504 | 0.531 | 0.285 | 0.396 |
| | PointWOLF | 0.919 | 0.870 | 0.405 | 1.111 | 0.915 | 1.032 | 1.121 | 0.688 | 0.815 |
| | PointWOLF+FAT (Ours) | 0.925 | **0.807** | 0.428 | 0.522 | 0.915 | 0.831 | 1.159 | 1.058 | 0.735 |
| | WOLFMix | 0.920 | 0.601 | 0.428 | 0.937 | 0.968 | 0.540 | 0.589 | 0.298 | 0.444 |
| | WOLFMix+FAT (Ours) | 0.930 | **0.537** | 0.530 | 0.418 | 1.138 | 0.460 | 0.527 | 0.281 | 0.404 |

Table 3: Quantitative results of FAT and its variants. FAT w/o low-frequency has a lower mCE for high-frequency corruptions such as "Jitter", while FAT w/o high-frequency has a lower mCE for low-frequency corruptions such as "scale". FAT w/o Advprop has a higher mCE but much worse OA. Compared with these methods, FAT achieves the lowest mCE.

| | Method | OA ↑ | mCE ↓ | Rotate | Jitter | Scale | Drop-G | Drop-L | Add-G | Add-L |
|---|---|---|---|---|---|---|---|---|---|---|
| PointNet | Vanilla Training | 0.907 | 1.422 | 1.902 | 0.642 | 1.266 | 0.500 | 1.072 | 2.980 | 1.593 |
| | FAT w/o low-frequency | 0.890 | 1.306 | 1.614 | 0.373 | 1.734 | 0.504 | 1.193 | 2.627 | 1.098 |
| | FAT w/o high-frequency | 0.906 | 1.317 | 1.702 | 0.519 | 1.234 | 0.452 | 1.043 | 2.851 | 1.415 |
| | FAT w/o frequency-division | 0.904 | 1.310 | 1.679 | 0.516 | 1.351 | 0.444 | 1.106 | 2.817 | 1.255 |
| | FAT w/o Advprop | 0.885 | 1.263 | 1.470 | 0.411 | 1.926 | 0.500 | 1.164 | 2.461 | 0.909 |
| | FAT | 0.902 | **1.237** | 1.553 | 0.370 | 1.606 | 0.448 | 1.097 | 2.583 | 1.004 |

## 4.4 Ablation study

In this section, we conduct ablation study among our proposed Frequency Adversarial Training (FAT), as well as FAT variants: FAT w/o low-frequency, FAT w/o high-frequency, FAT w/o frequency-division, and FAT w/o Advprop. FAT w/o low-frequency generates only high-frequency adversarial samples, while FAT w/o high-frequency generates only low-frequency adversarial samples. FAT w/o frequency-division randomly generates adversarial samples within a certain frequency range, without dividing the high and low frequency bands. FAT w/o Advprop does not use the AdvProp training framework [72]. We compare these methods in Table 3 based on mean corruption error (mCE), clean overall accuracy (OA), and corruption error (CE) measurements for each corruption type.

Compared with other methods, FAT w/o low-frequency has a lower mCE for high-frequency corruptions such as "Jitter", while FAT w/o high-frequency has a lower mCE for low-frequency corruptions such as "scale". As discussed in Sec. 3.3, this is because adversarial training on high/low frequencies reduces the high/low frequency sensitivity, thus improving robustness to high/low-frequency corruptions. The performance of FAT w/o frequency-division falls between FAT w/o low-frequency and FAT w/o high-frequency. Although FAT w/o Advprop has a better mCE, its clean overall accuracy (OA) is worse than the other methods due to mutual interference between samples from different distributions, which may cause potential performance degradation. Compared with these methods, FAT achieves the lowest mCE, showing the effectiveness of our algorithm. More experimental results can be found in Appendix B.

# 5 Conclusion

In this paper, we study the robustness of 3D point cloud recognition models under common corruptions. We focus on the frequency domain to analyze the underlying structure of point clouds and common corruptions. Through graph Fourier transform (GFT), we identify a correlation between the corruption robustness and the model sensitivity to different frequency bands. Motivated by the analysis, we propose Frequency Adversarial Training (FAT), an adversarial training method based on frequency-domain adversarial examples to improve the corruption robustness of 3D point cloud recognition models. Extensive experiments demonstrate that the proposed method significantly improves the corruption robustness of various point cloud models, and can be integrated with other data augmentation techniques to achieve the state-of-the-art performance.

**Limitation and broader impact.** A limitation of our proposed method is that it reduces the clean accuracy a bit, e.g., FAT reduces the clean accuracy of DGCNN by 0.1%, PointNet by 0.5%, PCT by 1.0%, and GDANet by 0.6%. This may be caused by the inherent trade-off between accuracy and robustness [88]. Additionally, despite the complexity in implementation, FAT does not affect the efficiency of model inference, ensuring unhindered deployment of well-trained models in practical applications. The robustness of 3D point cloud recognition under corruptions is a severe problem towards safe and reliable 3D perception. Our work proposes an effective method to solve this issue, which does not have any negative social impact.

## Acknowledgments

This work is supported by NKRDP grant No.2018YFA0704705, NSFC grants No.62276149 and No.12288201, and grant GJ0090202. Y. Dong is supported by the China National Postdoctoral Program for Innovative Talents. The authors thank anonymous referees for their valuable comments. L. Yu is supported by CAS Project for Young Scientists in Basic Research, Grant No.YSBR-040, ISCAS New Cultivation Project ISCAS-PYFX-202201, and ISCAS Basic Research ISCAS-JCZD-202302.

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

# A  Proof

**Theorem 1.** *If $f$ satisfies that: $f(x) \in [0, M]$ for all $x$, $|f(\mathcal{F}^{-1}(\mathcal{F}(x) + \alpha)) - f(x)| \leq \epsilon$ for all $x$ and $\|\alpha\|_p \leq \delta$, then for any distribution $P_o$ and $P_a$ satisfying that $Was^p(P_o, P_a) := (\inf_{u \in \Pi(P_o, P_a)} \mathbb{E}_{(x,z) \sim u}[\|\mathcal{F}(x) - \mathcal{F}(z)\|_p^p])^{1/p} \leq \eta$, where $\eta < \delta$, then, with probability $1 - \gamma$, we have:*

$$\mathbb{E}_{z \sim P_o}[f(z)] - \frac{1}{m} \sum_{i=1}^m f(x_i) \leq \epsilon \left( 1 - \frac{\eta^p}{\delta^p} - \sqrt{\frac{\ln(4/\gamma)}{2m}} \right) + \frac{\eta^p}{\delta^p} M + 4M \sqrt{\frac{\ln(4/\gamma)}{2m}}, \quad \text{(A.1)}$$

*where $\{x_i\}_{i=1}^m$ are i.i.d. samples from $P_a$.*

*Proof.* Assume $u$ is a joint distribution of $P_o$ and $P_a$, such that $(\mathbb{E}_{(x,z) \sim u}[\|\mathcal{F}(x) - \mathcal{F}(z)\|_p^p])^{1/p} \leq \eta$. Firstly, by Markov inequality, we have that:

$$\begin{aligned}
& P_{(x,z) \sim u}(\|\mathcal{F}(x) - \mathcal{F}(z)\|_p \geq \delta) \\
= \ & P_{(x,z) \sim u}(\|\mathcal{F}(x) - \mathcal{F}(z)\|_p^p \geq \delta^p) \\
\leq \ & \frac{\mathbb{E}_{(x,z) \sim u}[\|\mathcal{F}(x) - \mathcal{F}(z)\|_p^p]}{\delta^p} \\
\leq \ & \frac{\eta^p}{\delta^p}.
\end{aligned} \quad \text{(A.2)}$$

Then, we have that:

$$\mathbb{E}_{(x,z) \sim u}[I(\|\mathcal{F}(x) - \mathcal{F}(z)\|_p \leq \delta)] = P_{(x,z) \sim u}(\|\mathcal{F}(x) - \mathcal{F}(z)\|_p \leq \delta) \geq 1 - \frac{\eta^p}{\delta^p}. \quad \text{(A.3)}$$

Now, let $\{(x_i^u, z_i^u)\}_{i=1}^m$ be i.i.d. sampled from distribution $u$. Then, by Hoeffding inequality, we have that:

(1): with probability $1 - \gamma/4$, there are

$$\begin{aligned}
& \frac{1}{m} \sum_{i=1}^m I(\|\mathcal{F}(x_i^u) - \mathcal{F}(z_i^u)\|_p \leq \delta) \\
\geq \ & \mathbb{E}_{(x,z) \sim u}[I(\|\mathcal{F}(x) - \mathcal{F}(z)\|_p \leq \delta)] - \sqrt{\frac{ln(4/\gamma)}{2m}} \\
\geq \ & 1 - \frac{\eta^p}{\delta^p} - \sqrt{\frac{ln(4/\gamma)}{2m}},
\end{aligned} \quad \text{(A.4)}$$

which indicates that there are at least $m(1 - \frac{\eta^p}{\delta^p} - \sqrt{\frac{ln(4/\gamma)}{2m}})$ number of $i \in [m]$ makes that $\|\mathcal{F}(x_i^u) - \mathcal{F}(z_i^u)\|_p \leq \delta$;

(2): with probability $1 - \gamma/4$, there are

$$-\frac{1}{m} \sum_{i=1}^m f(z_i^u) + \mathbb{E}_{z \sim P_o}[f(z)] = -\frac{1}{m} \sum_{i=1}^m f(z_i^u) + \mathbb{E}_{(x,z) \sim u}[f(z)] \leq M \sqrt{\frac{ln(4/\gamma)}{2m}}; \quad \text{(A.5)}$$

(3): with probability $1 - \gamma/4$, there are

$$\frac{1}{m} \sum_{i=1}^m f(x_i^u) - \mathbb{E}_{x \sim P_a}[f(x)] = \frac{1}{m} \sum_{i=1}^m f(x_i^u) - \mathbb{E}_{(x,z) \sim u}[f(x)] \leq M \sqrt{\frac{ln(4/\gamma)}{2m}}; \quad \text{(A.6)}$$

Let $\{x_i\}_{i=1}^m$ are i.i.d. samples from distribution $P_a$, then, by Hoeffding inequality, we have that:

(4): with probability $1 - \gamma/4$, there are

$$-\frac{1}{m} \sum_{i=1}^m f(x_i) + \mathbb{E}_{x \sim P_a}[f(x)] \leq M \sqrt{\frac{ln(4/\gamma)}{2m}}; \quad \text{(A.7)}$$

So, with probability $1 - \gamma$ makes that (1), (2), (3) and (4) stand, at this times, we can estimate the $\mathbb{E}_{z \sim P_o}[f(z)] - \frac{1}{m} \sum_{i=1}^{m} f(x_i)$, there are:

$$
\begin{aligned}
& \mathbb{E}_{z \sim P_o}[f(z)] \\
\leq\ & \frac{1}{m} \sum_{i=1}^{m} f(z_i^u) + M\sqrt{\frac{ln(4/\gamma)}{2m}} \\
\leq\ & \frac{1}{m} \sum_{i=1}^{m} f(x_i^u) + |f(x_i^u) - f(z_i^u)| + M\sqrt{\frac{ln(4/\gamma)}{2m}} \\
\leq\ & \frac{1}{m} \sum_{i=1}^{m} f(x_i^u) + \epsilon I(\|\mathcal{F}(x_i^u) - \mathcal{F}(z_i^u)\|_p \leq \delta) + M I(\|\mathcal{F}(x_i^u) - \mathcal{F}(z_i^u)\|_p > \delta) + M\sqrt{\frac{ln(4/\gamma)}{2m}} \\
\leq\ & \frac{1}{m} \sum_{i=1}^{m} f(x_i^u) + \epsilon(1 - \frac{\eta^p}{\delta^p} - \sqrt{\frac{ln(4/\gamma)}{2m}}) + (\frac{\eta^p}{\delta^p} + \sqrt{\frac{ln(4/\gamma)}{2m}})M + M\sqrt{\frac{ln(4/\gamma)}{2m}} \\
\leq\ & \mathbb{E}_{x \sim P_a}[f(x)] + \epsilon(1 - \frac{\eta^p}{\delta^p} - \sqrt{\frac{ln(4/\gamma)}{2m}}) + (\frac{\eta^p}{\delta^p} + \sqrt{\frac{ln(4/\gamma)}{2m}})M + 2M\sqrt{\frac{ln(4/\gamma)}{2m}} \\
\leq\ & \frac{1}{m} \sum_{i=1}^{m} f(x_i) + \epsilon(1 - \frac{\eta^p}{\delta^p} - \sqrt{\frac{ln(4/\gamma)}{2m}}) + (\frac{\eta^p}{\delta^p} + \sqrt{\frac{ln(4/\gamma)}{2m}})M + 3M\sqrt{\frac{ln(4/\gamma)}{2m}} \\
=\ & \frac{1}{m} \sum_{i=1}^{m} f(x_i) + \epsilon(1 - \frac{\eta^p}{\delta^p} - \sqrt{\frac{ln(4/\gamma)}{2m}}) + M\frac{\eta^p}{\delta^p} + 4M\sqrt{\frac{ln(4/\gamma)}{2m}}.
\end{aligned}
$$

(A.8)

We get our conclusion. $\qquad \square$

Our generalization bound can also be extended to Lipschitz neural networks, which are a class of networks with global Lipschitz constants [11, 87].

**Corollary A.1.** *If $f$ satisfies that: $f(x) \in [0, M]$ for all $x \in [0, 1]^n$, $|f(\mathcal{F}^{-1}(\mathcal{F}(x) + \alpha)) - f(x)| \leq \epsilon \|\alpha\|_p$ for all $x \in [0, 1]^n$ and $\alpha$, then for any distribution $P_o$ and $P_a$ in $[0, 1]^n$ satisfying that $Was^p(P_o, P_a) := (\inf_{u \in \Pi(P_o, P_a)} \mathbb{E}_{(x,z) \sim u}[\|\mathcal{F}(x) - \mathcal{F}(z)\|_p^p])^{1/p} \leq \eta$, then, with probability $1 - \gamma$, we have:*

$$
\mathbb{E}_{z \sim P_o}[f(z)] - \frac{1}{m} \sum_{i=1}^{m} f(x_i) \leq \epsilon(\eta + v\eta\sqrt{\frac{ln(4/\gamma)}{2m}}) + \frac{M}{v^p} + 3M\sqrt{\frac{ln(4/\gamma)}{2m}}, \qquad (A.9)
$$

*where $\{x_i\}_{i=1}^{m}$ are i.i.d. samples from $P_a$, $v$ is any real number greater than 1.*

*Proof.* Assuming $u$ is a joint distribution of $P_o$ and $P_a$, and makes that $(\mathbb{E}_{(\mathcal{F}(x), \mathcal{F}(z)) \sim u}[\|x - z\|_p^p])^{1/p} \leq \eta$. Firstly, by Markov inequality, we have that:

$$
\begin{aligned}
& P_{(x,z) \sim u}(\|\mathcal{F}(x) - \mathcal{F}(z)\|_p \geq v\eta) \\
=\ & P_{(x,z) \sim u}(\|\mathcal{F}(x) - \mathcal{F}(z)\|_p^p \geq (v\eta)^p) \\
\leq\ & \frac{\mathbb{E}_{(x,z) \sim u}[\|\mathcal{F}(x) - \mathcal{F}(z)\|_p^p]}{(v\eta)^p} \\
\leq\ & \frac{\eta^p}{(v\eta)^p} = (1/v)^p.
\end{aligned}
$$

(A.10)

Then, we have that:

$$
\mathbb{E}_{(x,z) \sim u}(I(\|\mathcal{F}(x) - \mathcal{F}(z)\|_p \leq v\eta)) = P_{(x,z) \sim u}(\|\mathcal{F}(x) - \mathcal{F}(z)\|_p \leq v\eta) \geq 1 - \frac{1}{v^p}. \quad (A.11)
$$

Now, let $\{(x_i, z_i)\}_{i=1}^{m}$ are i.i.d. samples from distribution $u$. Then, by Hoeffding inequality, we have that:

(1): with probability $1 - \gamma/4$, there are

$$
\begin{aligned}
& \frac{1}{m} \sum_{i=1}^{m} I(\|\mathcal{F}(x_i^u) - \mathcal{F}(z_i^u)\|_p \leq v\eta)\|\mathcal{F}(x_i^u) - \mathcal{F}(z_i^u)\|_p \\
\leq\ & \mathbb{E}_{(x,z) \sim u}[I(\|\mathcal{F}(x_i^u) - \mathcal{F}(z_i^u)\|_p \\
\leq\ & v\eta)\|\mathcal{F}(x) - \mathcal{F}(z)\|_p] + v\eta\sqrt{\frac{ln(4/\gamma)}{2m}} \\
\leq\ & (\mathbb{E}_{(x,z) \sim u}[\|\mathcal{F}(x) - \mathcal{F}(z)\|_p^p])^{1/p} + v\eta\sqrt{\frac{ln(4/\gamma)}{2m}} \\
=\ & \eta + v\eta\sqrt{\frac{ln(4/\gamma)}{2m}};
\end{aligned}
$$

(A.12)

(2): with probability $1 - \gamma/4$, there are

$$
-\frac{1}{m} \sum_{i=1}^{m} f(z_i^u) + \mathbb{E}_{z \sim P_o}[f(z)]| = -\frac{1}{m} \sum_{i=1}^{m} f(z_i^u) + \mathbb{E}_{(x,z) \sim u}[f(z)] \leq M\sqrt{\frac{ln(4/\gamma)}{2m}}; \quad (A.13)
$$

(3): with probability $1 - \gamma/4$, there are

$$\frac{1}{m}\sum_{i=1}^{m} f(x_i^u) - \mathbb{E}_{x \sim P_a}[f(x)] = \frac{1}{m}\sum_{i=1}^{m} f(x_i^u) - \mathbb{E}_{(x,z)\sim u}[f(x)] \le M\sqrt{\frac{ln(4/\gamma)}{2m}}; \qquad \text{(A.14)}$$

Let $\{x_i\}_{i=1}^{m}$ are i.i.d. samples from distribution $P_a$, then, by Hoeffding inequality, we have that:

(4): with probability $1 - \gamma/4$, there are

$$-\frac{1}{m}\sum_{i=1}^{m} f(x_i) + \mathbb{E}_{x \sim P_a}[f(x)] \le M\sqrt{\frac{ln(4/\gamma)}{2m}}; \qquad \text{(A.15)}$$

So, with probability $1 - \gamma$ makes that (1), (2), (3) and (4) stand, at this times, we can estimate the $\mathbb{E}_{z \sim P_o}[f(z)] - \frac{1}{m}\sum_{i=1}^{m} f(x_i)$, there are:

$$
\begin{aligned}
& \mathbb{E}_{z \sim P_o}[f(z)] \\
\le\ & \tfrac{1}{m}\sum_{i=1}^{m} f(z_i^u) + M\sqrt{\tfrac{ln(4/\gamma)}{2m}} \\
\le\ & \tfrac{1}{m}\sum_{i=1}^{m} f(x_i^u) + |f(x_i^u) - f(z_i^u)| + M\sqrt{\tfrac{ln(4/\gamma)}{2m}} \\
\le\ & \tfrac{1}{m}\sum_{i=1}^{m} f(x_i^u) + \epsilon\|\mathcal{F}(x_i^u) - \mathcal{F}(z_i^u)\|_p I(\|\mathcal{F}(x_i^u) - \mathcal{F}(z_i^u)\|_p \le v\eta) \\
& + MI(\|\mathcal{F}(x_i^u) - \mathcal{F}(z_i^u)\|_p > v\eta) + M\sqrt{\tfrac{ln(4/\gamma)}{2m}} \\
\le\ & \tfrac{1}{m}\sum_{i=1}^{m} f(x_i^u) + \epsilon(\eta + v\eta\sqrt{\tfrac{ln(4/\gamma)}{2m}}) + \tfrac{M}{v^p} + M\sqrt{\tfrac{ln(4/\gamma)}{2m}} \\
\le\ & \mathbb{E}_{x \sim P_a}[f(x)] + \epsilon(\eta + v\eta\sqrt{\tfrac{ln(4/\gamma)}{2m}}) + \tfrac{M}{v^p} + 2M\sqrt{\tfrac{ln(4/\gamma)}{2m}} \\
\le\ & \tfrac{1}{m}\sum_{i=1}^{m} f(x_i) + \epsilon(\eta + v\eta\sqrt{\tfrac{ln(4/\gamma)}{2m}}) + \tfrac{M}{v^p} + 3M\sqrt{\tfrac{ln(4/\gamma)}{2m}}.
\end{aligned}
\qquad \text{(A.16)}
$$

We get our conclusion. $\qquad\qquad\square$

# B   Supplementary experimental results

In this section, we provide more experimental results. All of the experiments are conducted on NVIDIA Tesla V100 GPUs.

## B.1   The performance in terms of mOA and RmCE

In this section, we present full results for corruption overall accuracy (mOA) and relative mCE (RmCE) [41]. The mOA is computed as the average OA over all corruptions. The RmCE quantifies the performance drop compared to a clean test set. We adopt the official baseline DGCNN and

Table B.1: Quantitative results of vanilla training, adversarial training and our proposed Frequency Adversarial Training (FAT) on the ModelNet-C test set. Our proposed FAT outperforms all other methods in terms of corruption overall accuracy (mOA), which demonstrates the effectiveness of FAT for improving corruption robustness.

|  | Method | OA ↑ | mOA ↑ | Rotate | Jitter | Scale | Drop-G | Drop-L | Add-G | Add-L |
|---|---|---|---|---|---|---|---|---|---|---|
| DGCNN | Vanilla Training | 0.926 | 0.764 | 0.785 | 0.684 | 0.906 | 0.752 | 0.793 | 0.705 | 0.725 |
|  | Adv Training | 0.925 | 0.790 | 0.781 | 0.816 | 0.902 | 0.753 | 0.772 | 0.743 | 0.761 |
|  | FAT (Ours) | 0.925 | **0.815** | 0.807 | 0.857 | 0.907 | 0.769 | 0.799 | 0.772 | 0.791 |
| PointNet | Vanilla Training | 0.907 | 0.658 | 0.591 | 0.797 | 0.881 | 0.876 | 0.778 | 0.121 | 0.562 |
|  | Adv Training | 0.904 | 0.673 | 0.602 | 0.822 | 0.879 | 0.889 | 0.777 | 0.148 | 0.591 |
|  | FAT (Ours) | 0.902 | **0.717** | 0.666 | 0.883 | 0.849 | 0.889 | 0.773 | 0.238 | 0.724 |
| PCT | Vanilla Training | 0.930 | 0.781 | 0.776 | 0.725 | 0.918 | 0.869 | 0.793 | 0.770 | 0.619 |
|  | Adv Training | 0.919 | 0.778 | 0.776 | 0.877 | 0.899 | 0.774 | 0.753 | 0.673 | 0.693 |
|  | FAT (Ours) | 0.920 | **0.798** | 0.783 | 0.891 | 0.898 | 0.791 | 0.744 | 0.731 | 0.747 |
| GDANet | Vanilla Training | 0.934 | 0.789 | 0.789 | 0.735 | 0.922 | 0.803 | 0.815 | 0.743 | 0.715 |
|  | Adv Training | 0.926 | 0.781 | 0.761 | 0.840 | 0.903 | 0.770 | 0.764 | 0.686 | 0.742 |
|  | FAT (Ours) | 0.928 | **0.810** | 0.749 | 0.871 | 0.913 | 0.803 | 0.770 | 0.807 | 0.756 |

Table B.2: Quantitative results of vanilla training, adversarial training and our proposed Frequency Adversarial Training (FAT) on the ModelNet-C test set. Our proposed FAT outperforms all other methods in terms of relative mCE (RmCE), which demonstrates the effectiveness of FAT for improving corruption robustness.

| | Method | OA ↑ | RmCE ↓ | Rotate | Jitter | Scale | Drop-G | Drop-L | Add-G | Add-L |
|---|---|---|---|---|---|---|---|---|---|---|
| DGCNN | Vanilla Training | 0.926 | 1.000 | 1.000 | 1.000 | 1.000 | 1.000 | 1.000 | 1.000 | 1.000 |
| | Adv Training | 0.925 | 0.914 | 1.021 | 0.450 | 1.150 | 0.989 | 1.150 | 0.824 | 0.816 |
| | FAT (Ours) | 0.925 | **0.746** | 0.837 | 0.281 | 0.899 | 0.897 | 0.947 | 0.692 | 0.667 |
| PointNet | Vanilla Training | 0.907 | 1.488 | 2.241 | 0.455 | 1.300 | 0.178 | 0.970 | 3.557 | 1.716 |
| | Adv Training | 0.904 | 1.393 | 2.142 | 0.339 | 1.250 | 0.086 | 0.955 | 3.421 | 1.557 |
| | FAT (Ours) | 0.902 | **1.334** | 1.674 | 0.079 | 2.650 | 0.075 | 0.970 | 3.005 | 0.886 |
| PCT | Vanilla Training | 0.930 | 0.884 | 1.092 | 0.847 | 0.600 | 0.351 | 1.030 | 0.724 | 1.547 |
| | Adv Training | 0.919 | 0.929 | 1.014 | 0.174 | 1.000 | 0.833 | 1.248 | 1.113 | 1.124 |
| | FAT (Ours) | 0.920 | **0.853** | 0.972 | 0.120 | 1.098 | 0.741 | 1.323 | 0.855 | 0.861 |
| GDANet | Vanilla Training | 0.934 | 0.865 | 0.753 | 0.822 | 0.600 | 0.895 | 0.864 | 1.090 | 1.028 |
| | Adv Training | 0.926 | 0.970 | 1.170 | 0.355 | 1.150 | 0.897 | 1.218 | 1.086 | 0.915 |
| | FAT (Ours) | 0.928 | **0.795** | 1.270 | 0.236 | 0.750 | 0.718 | 1.188 | 0.548 | 0.856 |

Table B.3: Quantitative results of vanilla training and our proposed Frequency Adversarial Training (FAT) on the ModelNet40-C test set. Our proposed FAT outperforms other methods in terms of mCE, which demonstrates the effectiveness of FAT for improving corruption robustness.

| | Method | OA ↑ | mCE ↓ | Uni. | Gauss. | Impluse | Upsamp. | Back. | Occlu. | LiDAR | Den.Inc. | Den.Dec. | Cutout | Rotate | Shear | FFD | RBF | Inv.RBF |
|---|---|---|---|---|---|---|---|---|---|---|---|---|---|---|---|---|---|---|
| DGCNN | VT | 0.926 | 1.000 | 1.000 | 1.000 | 1.000 | 1.000 | 1.000 | 1.000 | 1.000 | 1.000 | 1.000 | 1.000 | 1.000 | 1.000 | 1.000 | 1.000 | 1.000 |
| | FAT | 0.925 | **0.782** | 0.687 | 0.601 | 0.551 | 0.580 | 0.764 | 0.952 | 0.912 | 0.881 | 0.919 | 0.863 | 0.808 | 0.824 | 0.817 | 0.787 | 0.785 |
| PointNet | VT | 0.907 | 1.157 | 0.850 | 0.871 | 1.168 | 0.733 | 1.763 | 0.883 | 0.677 | 0.739 | 0.671 | 0.778 | 1.930 | 2.105 | 1.626 | 1.277 | 1.274 |
| | FAT | 0.902 | **1.009** | 0.763 | 0.731 | 0.833 | 0.631 | 1.689 | 0.904 | 0.694 | 0.710 | 0.633 | 0.730 | 1.688 | 1.807 | 1.328 | 1.012 | 0.979 |
| PCT | VT | 0.929 | 0.959 | 0.828 | 0.839 | 1.568 | 0.908 | 1.091 | 0.956 | 0.947 | 0.837 | 0.829 | 0.942 | 0.949 | 0.956 | 0.943 | 0.892 | 0.898 |
| | FAT | 0.920 | **0.699** | 0.584 | 0.535 | 0.519 | 0.553 | 0.604 | 0.953 | 0.739 | 0.653 | 0.757 | 0.753 | 0.872 | 0.858 | 0.814 | 0.647 | 0.644 |
| GDANet | VT | 0.934 | 0.869 | 0.950 | 0.995 | 0.912 | 0.864 | 0.801 | 0.987 | 0.829 | 0.698 | 0.690 | 0.777 | 0.947 | 0.914 | 0.883 | 0.900 | 0.885 |
| | FAT | 0.928 | **0.764** | 0.645 | 0.604 | 0.502 | 0.579 | 0.508 | 0.998 | 0.817 | 0.776 | 0.805 | 0.819 | 1.029 | 0.979 | 0.888 | 0.758 | 0.753 |

initially calculate the relative corruption error (RCE) for a given corruption type $i$ by averaging over 5 severity levels: $\mathrm{RCE}_i = \frac{\sum_{l=1}^{5}(\mathrm{OA}_{\mathrm{clean}}-\mathrm{OA}_{i,l})}{\sum_{l=1}^{5}(\mathrm{OA}_{\mathrm{clean}}^{\mathrm{DGCNN}}-\mathrm{OA}_{i,l}^{\mathrm{DGCNN}})}$, where $\mathrm{OA}_{\mathrm{clean}}$ is the overall accuracy on the clean test set. Subsequently, we compute the relative mean corruption error (RmCE) by averaging over the 7 corruption types: $\mathrm{RmCE} = \frac{1}{N}\sum_{i=1}^{N}\mathrm{RCE}_i$. In Tables B.1 and B.2, we compare different methods based on the mOA and RmCE metrics, confirming that our proposed FAT enhances the model's out-of-distribution generalization ability.

## B.2 The performance on the ModelNet40-C

In this section, we evaluate the corruption robustness on ModelNet40-C [51]. The ModelNet40-C dataset is specifically designed to assess the network robustness against prevalent point cloud corruptions. It consists of 15 different corruption types, including "Uniform", "Gaussian", "Impulse", "Upsampling", "Background", "Occlusion", "LiDAR", "Local Density Inc", "Local Density Dec", "Cutout", "Rotation", "Shear", "FFD", "RBF", and "Inv RBF". Each type of corruption has 5 severity

Table B.4: Quantitative results of the performance of FAT when integrated with data augmentation strategy in terms of mCE and ER$_{\mathrm{cor}}$ on the ModelNet40-C test set. In previous studies [51], PCT with CutMix-R achieves the best robustness with the 0.635 mCE and 0.163 ER$_{\mathrm{cor}}$. However, training GDANet with the combination of our proposed FAT with WOLFMix achieves the new state-of-the-art performance, with the impressive 0.555 mCE and **0.147 ER$_{\mathrm{cor}}$**.

| Method | OA ↑ | mCE ↓ | Noise | Density | Trans. | ER$_{\mathrm{cor}}$ ↓ | Noise | Density | Trans. |
|---|---|---|---|---|---|---|---|---|---|
| PCT+CutMix-R | 0.928 | 0.635 | 0.469 | 0.669 | 0.766 | 0.163 | 0.105 | 0.271 | 0.112 |
| PCT+WOLFMix | 0.922 | 0.627 | 0.580 | 0.673 | 0.629 | 0.158 | 0.118 | 0.267 | 0.090 |
| PCT+WOLFMix+FAT (Ours) | 0.920 | **0.583** | 0.454 | 0.636 | 0.658 | **0.151** | 0.097 | 0.261 | 0.094 |
| GDANet+WOLFMix | 0.920 | 0.669 | 0.709 | 0.685 | 0.612 | 0.171 | 0.137 | 0.289 | 0.087 |
| GDANet+WOLFMix+FAT (Ours) | 0.930 | **0.555** | 0.450 | 0.627 | 0.588 | **0.147** | 0.094 | 0.263 | 0.084 |

Table B.5: Quantitative results of vanilla training, adversarial training, DUP Defense and our proposed Frequency Adversarial Training (FAT) on the ScanObjectNN-C test set. Our proposed FAT outperforms all other methods in terms of mean corruption error (mCE), which demonstrates the effectiveness of FAT for improving corruption robustness.

| | Method | OA ↑ | mCE ↓ | Rotate | Jitter | Scale | Drop-G | Drop-L | Add-G | Add-L |
|---|---|---|---|---|---|---|---|---|---|---|
| DGCNN | Vanilla Training | 0.858 | 1.000 | 1.000 | 1.000 | 1.000 | 1.000 | 1.000 | 1.000 | 1.000 |
| | Adv Training | 0.843 | 1.062 | 1.146 | 0.778 | 1.097 | 0.873 | 0.960 | 1.185 | 1.396 |
| | DUP Defense | 0.832 | 1.029 | 1.195 | 0.833 | 1.157 | 1.197 | 1.214 | 0.773 | 0.834 |
| | FAT (Ours) | 0.856 | **0.933** | 0.968 | 0.815 | 0.959 | 0.852 | 0.950 | 0.959 | 1.026 |
| PointNet | Vanilla Training | 0.739 | 1.354 | 1.610 | 0.884 | 1.427 | 0.786 | 1.264 | 1.487 | 2.022 |
| | Adv Training | 0.725 | 1.334 | 1.532 | 0.844 | 1.403 | 0.873 | 1.333 | 1.474 | 1.881 |
| | DUP Defense | 0.712 | 1.348 | 1.717 | 0.825 | 1.555 | 1.157 | 1.480 | 0.829 | 1.875 |
| | FAT (Ours) | 0.734 | **1.254** | 1.393 | 0.796 | 1.465 | 0.844 | 1.264 | 1.259 | 1.759 |
| PointNext | Vanilla Training | 0.873 | 0.921 | 0.995 | 1.079 | 0.803 | 0.807 | 0.942 | 0.944 | 0.875 |
| | Adv Training | 0.870 | 0.901 | 0.991 | 1.027 | 0.803 | 0.833 | 0.929 | 0.912 | 0.809 |
| | DUP Defense | 0.859 | 0.901 | 0.980 | 1.046 | 0.826 | 0.748 | 0.973 | 0.923 | 0.809 |
| | FAT (Ours) | 0.875 | **0.877** | 0.998 | 0.916 | 0.791 | 0.786 | 0.867 | 0.938 | 0.840 |

Table B.6: Quantitative results of vanilla training, adversarial training, DUP Defense and our proposed Frequency Adversarial Training (FAT) on the PointNeXt on ModelNet-C. Our proposed FAT outperforms all other methods in terms of mean corruption error (mCE), which demonstrates the effectiveness of FAT for improving corruption robustness.

| | Method | OA ↑ | mCE ↓ | Rotate | Jitter | Scale | Drop-G | Drop-L | Add-G | Add-L |
|---|---|---|---|---|---|---|---|---|---|---|
| PointNeXt | Vanilla Training | 0.932 | 0.856 | 1.460 | 1.297 | 0.904 | 0.847 | 0.957 | 0.251 | 0.276 |
| | Adv Training | 0.924 | 0.834 | 1.593 | 0.716 | 1.025 | 0.876 | 1.144 | 0.230 | 0.251 |
| | DUP Defense | 0.919 | 0.840 | 1.461 | 0.838 | 1.192 | 0.715 | 1.188 | 0.224 | 0.262 |
| | FAT (Ours) | 0.930 | **0.781** | 1.412 | 0.692 | 0.986 | 0.827 | 1.082 | 0.230 | 0.241 |

levels. In Table B.3, we compare different methods on the ModelNet40-C test set, confirming that our proposed FAT enhances the OOD generalization ability of the model.

In Table B.4, we show the performance of FAT when integrated with data augmentation strategy in terms of mCE and $ER_{cor}$. In previous studies [51], PCT with CutMix-R achieves the best robustness with the 0.635 mCE and 0.163 $ER_{cor}$. However, training GDANet with the combination of our proposed FAT with WOLFMix achieves a new state-of-the-art performance, with the impressive 0.555 mCE and 0.147 $ER_{cor}$.

## B.3 The performance on the ScanObjectNN-C

We further conduct experiments on the ScanObjectNN dataset, which is collected by LiDAR sensors and represents more realistic conditions under real-world scenarios [2, 58, 96, 94]. The experimental settings and evaluation metrics on ScanObjectNN-C [62] are consistent with those on ModelNet-C. The results are shown in Table B.5. It can be seen that our FAT generally leads to lower mCE on ScanObjectNN-C. The experimental results on ScanObjectNN-C further validate the generalizability and applicability of our FAT under real-world conditions.

## B.4 The performance on the PointNeXt

We further conduct experiments for the updated point cloud model PointNeXt [37]. The results in Table B.5 and Table B.6 demonstrate that FAT achieves consistent performance on the advanced network architecture PointNeXt, similar to observations on PointNet and more. Our FAT outperforms all other methods in terms of mCE. This indicates that FAT's performance is largely independent of the underlying model architecture, making it applicable to both traditional and modern networks.

## B.5 The performance in combination with AdaptPoint

We further conduct experiments for comparison with AdaptPoint [62] on ScanObjectNN-C. Adapt-Point follows the official experimental settings. The results are shown in Table B.7. It is evident that incorporating FAT achieves a lower mCE, indicating its superiority.

Table B.7: Quantitative results of combining FAT with AdaptPoint on ScanObjectNN-C. Compared with single AdaptPoint, the combination of FAT and AdaptPoint achieves a better mCE.

| | Method | OA ↑ | **mCE** ↓ | Rotate | Jitter | Scale | Drop-G | Drop-L | Add-G | Add-L |
|---|---|---|---|---|---|---|---|---|---|---|
| PointNet | AdaptPoint | 0.743 | 1.256 | 1.359 | 0.875 | 1.519 | 0.676 | 1.112 | 1.448 | 1.804 |
| | AdaptPoint+FAT (Ours) | 0.744 | **1.196** | 1.370 | 0.823 | 1.446 | 0.690 | 1.125 | 1.220 | 1.701 |
| PointNext | AdaptPoint | 0.885 | 0.783 | 0.767 | 1.030 | 0.810 | 0.508 | 0.628 | 0.911 | 0.824 |
| | AdaptPoint+FAT (Ours) | 0.885 | **0.761** | 0.748 | 0.948 | 0.833 | 0.521 | 0.648 | 0.829 | 0.802 |

