# OpenReview forum: "Improving Robustness of 3D Point Cloud Recognition from a Fourier Perspective"
_NeurIPS.cc/2024/Conference — NeurIPS 2024 poster_

### Official Review · Reviewer_ELyT · 2024-07-04

**Soundness:** 3
**Presentation:** 3
**Contribution:** 3
**Rating:** 6
**Confidence:** 3

**Summary:**

This work introduces a method called Frequency Adversarial Training (FAT) to improve the robustness of 3D point cloud recognition models and examines the robustness of models under 3D point cloud corruptions, including analysis on the power of different corruption effects in the frequency domain.  FAT generates adversarial samples by adding perturbations to the frequency representations of point cloud data. The authors also provide a theoretical analysis demonstrating the effectiveness of FAT in improving  OOD generalization performance of models.

**Strengths:**

1. The paper is well-structured and organized, making it easy to understand and follow.
2. Introduces a novel concept of frequency augmentation for 3D point cloud data, while previous works  are mainly mix-based and deformation-based on original point cloud data rather than their frequency representation.
3. The proposed method considers distribution shifts caused by adversarial perturbations and thus uses separate batch normalization layers,    improving model robustness against both low-frequency and high-frequency corruptions. The authors also provide an ablation study demonstrating the effectiveness of each component in improving robustness.
4. The authors provide theoretical analysis showing that adversarial robustness in the frequency domain enhances real-world corruption robustness, and a frequency sensitivity measurement which offers valuable insights for analyzing model robustness from a frequency perspective.

**Weaknesses:**

1. There is confusion in the caption of Figure 3, which suggests the Jacobian matrix for an input point cloud. According to the context, it should be the Jacobian matrix of the model regarding the input point cloud.
2.  As FAT is based on adversarial training, it is expected to have comparisons with other adversarial training techniques rather than only mixing- and deformation-based approaches.
3. Minors:
*The figures are hard to read as the font size is too small.
*Missing space before the fourth sentence in the caption of Figure 3.

**Questions:**

1. Adversarial augmentation approaches may require substantial computational resources, leading to inefficiency. How much computational resources does FAT require regarding computational time, memory usage, etc., and how does it compare with other augmentation approaches.

**Limitations:**

The authors address the limitation of their approach in term of the degraded standard accuracy. There might be a trade-off between corruption robustness and standard performance, left for future investigation.

---

> ### Author Rebuttal · Authors · 2024-08-07
>
> Thank you for appreciating our new contributions as well as providing the valuable feedback. Below we address the detailed comments, and hope that you can find our response satisfactory.
>
> ***Question 1: There is confusion in the caption of Figure 3, which suggests the Jacobian matrix for an input point cloud. According to the context, it should be the Jacobian matrix of the model regarding the input point cloud.***
>
> Thanks for pointing out this issue. We will clarify this in the revision.
>
> ***Question 2: As FAT is based on adversarial training, it is expected to have comparisons with other adversarial training techniques rather than only mixing- and deformation-based approaches.***
>
> We have already compared our proposed FAT with adversarial training technique on the ModelNet-C test set in Table 1 in Sec. 4.2. Our FAT outperforms all other methods in terms of mCE (mean corruption error).
>
> ***Question 3: Minors: The figures are hard to read as the font size is too small. Missing space before the fourth sentence in the caption of Figure 3.***
>
> Thanks for pointing out these issues. We will correct them in the revision.
>
> ***Question 4: How much computational resources does FAT require regarding computational time, memory usage, etc., and how does it compare with other augmentation approaches.***
>
> The computational cost of our FAT implementation primarily involves generating high-frequency and low-frequency adversarial examples.
> Compared to standard adversarial training and other data augmentations, FAT incurs approximately $1.7 \sim 3.2$ times higher computational costs and about 3 times more memory usage.
> Despite this limitation, such an increase in computational overhead is deemed acceptable for offline training scenarios.
> Moreover, our approach would not affect the efficiency of model inference, ensuring unhindered deployment of well-trained models in practical applications.
> We will add the discussion in the revision.

---

> > ### Comment · Reviewer_ELyT · 2024-08-11
> >
> > Thank you for the responses. I've updated the rating.

---

> > > ### Author Response · Authors · 2024-08-14
> > > **Thank you for increasing the score**
> > >
> > > Dear Reviewer ELyT,
> > >
> > > Thank you very much for increasing the score! We are glad to know that our response has addressed your concerns. We really appreciate your valuable comments and appreciation of our contributions. We will further improve the paper in the final.
> > >
> > > Best regards, Authors

---

### Official Review · Reviewer_wa2k · 2024-07-08

**Soundness:** 3
**Presentation:** 3
**Contribution:** 3
**Rating:** 5
**Confidence:** 5

**Summary:**

This paper introduces Frequency Adversarial Training (FAT), leveraging Graph Fourier Transform (GFT) to enhance robustness against point cloud corruptions by training models with frequency-domain adversarial examples, demonstrating significant improvements in robustness across various architectures through extensive experiments.

**Strengths:**

1.	Innovative Approach: The use of the frequency domain for analyzing and improving the robustness of point cloud recognition models is novel and well-motivated. The application of GFT to understand the impact of corruptions on different frequency bands is a significant contribution.
2.	Comprehensive Evaluation: The paper provides a thorough evaluation of the proposed method, comparing it with existing approaches and demonstrating its effectiveness across multiple models and datasets.
3.	Theoretical and Empirical Validation: The authors provide both theoretical analysis and empirical evidence to support the effectiveness of FAT, strengthening the credibility of their claims.
4.	Practical Relevance: Improving the robustness of 3D point cloud recognition models is highly relevant for safety-critical applications such as autonomous driving and robotics.
5.	Writing quality: This paper is written and organized well

**Weaknesses:**

1.	Complexity of Implementation: The proposed FAT method involves several complex steps, including the generation of high-frequency and low-frequency adversarial examples and the use of multiple batch normalizations. This complexity might hinder the adoption of the method in practical scenarios.
2.	Limited Impact on Clean Accuracy: The paper mentions a slight reduction in clean accuracy when using FAT, which might be a concern for applications where both robustness and accuracy are critical.
3.	Generality of Results: While the experiments demonstrate the effectiveness of FAT on several models and datasets, additional experiments on more diverse datasets and real-world scenarios could further validate the generalizability of the approach, e.g.ShapeNet-C[1] and ScanObjectNN-C[2]
4.	Lack of Updated Baselines in Experiments: In Table 1, the paper lacks experiments comparing some updated point cloud baselines, like RPC[3] and PointNeXt[4], which could provide a more comprehensive evaluation of the proposed method.
5.	Missing Comparison with Updated Augmentation Methods: In Table 2, the paper lacks comparison with updated augmentation methods such as AdaptPoint[2], which could highlight the relative performance of FAT against newer augmentation techniques.

[1]: PointCloud-C: Benchmarking and Analyzing Point Cloud Perception Robustness under Corruptions
[2]: Sample-adaptive Augmentation for Point Cloud Recognition Against Real-world Corruptions
[3]: Benchmarking and Analyzing Point Cloud Classification under Corruptions
[4]: PointNeXt: Revisiting PointNet++ with Improved Training and Scaling Strategies

**Questions:**

Refer to the Weakness. I will improve my rates when the weaknesses are solved.

**Limitations:**

Refer to the Weakness.

---

> ### Author Rebuttal · Authors · 2024-08-07
>
> Thank you for acknowledging the novelty of our paper as well as providing the valuable feedback. Below we address the detailed comments, and hope that you can find our response satisfactory.
>
> ***Question 1: Complexity of Implementation might hinder the adoption of the method in practical scenarios.***
>
> The computational cost of our FAT implementation primarily involves generating high-frequency and low-frequency adversarial examples.
> Compared to standard adversarial training and other data augmentations, FAT incurs approximately $1.7 \sim 3.2$ times higher computational costs.
> Despite this limitation, such an increase in computational overhead is deemed acceptable for offline training scenarios.
> Moreover, our approach would not affect the efficiency of model inference, ensuring unhindered deployment of well-trained models in practical applications.
> We will add the discussion in the revision.
>
> ***Question 2: The paper mentions a slight reduction in clean accuracy when using FAT, which might be a concern for applications.***
>
> In our limitations section, we have already noted that while FAT slightly reduces clean accuracy, the magnitude of this reduction is minimal and deemed acceptable. This could be attributed to the inherent trade-off between accuracy and robustness [1].
> Moreover, as shown in Table 2, existing data augmentation methods decrease clean accuracy by an average of 1.03\%, whereas our FAT exhibits a smaller average reduction of 0.55\% in clean accuracy.
> Additionally, Table 2 demonstrates that incorporating FAT results in an average improvement of 0.33\% in clean accuracy, indicating that FAT's impact on clean accuracy is not universally negative.
>
> ***Question 3: The paper would be more convincing by additional experiments on more diverse datasets and real-world scenarios.***
>
> Thanks for the valuable suggestion. We further conduct experiments on the KITTI and ScanObjectNN datasets, both collected by LiDAR sensors.
> ***Due to space constraints, detailed experimental results and analysis are presented in the Global Rebuttal.***
> (See more results on ScanObjectNN-C in response to Question 5).
> Experiments on these two real-world datasets [2][3] further validate the superiority and practicality of our FAT.
>
> ***Question 4: The paper lacks experiments comparing some updated point cloud baselines, which could provide a more comprehensive evaluation of the proposed method.***
>
> Thanks for the valuable suggestion. We further conduct experiments for PointNeXt [4] on ModelNet-C and ScanObjectNN-C below. (See more results on PointNeXt in response to Question 5.)
>
> |ModelNet-C|Method|OA|mCE|Rotate|Jitter|Scale|Drop-G|Drop-L|Add-G|Add-L
> |:-:|:-:|:-:|:-:|:-:|:-:|:-:|:-:|:-:|:-:|:-:|
> |PointNext|Vanilla Training|0.932|0.856|1.460|1.297|0.904|0.847|0.957|0.251|0.276|
> ||Adv Training|0.924|0.834|1.593|0.716|1.025|0.876|1.144|0.230|0.251|
> ||DUP Defense|0.919|0.840|1.461|0.838|1.192|0.715|1.188|0.224|0.262|
> ||FAT (Ours)|0.930|**0.781**|1.412|0.692|0.986|0.827|1.082|0.230|0.241|
>
> |ScanObjectNN-C|Method|OA|mCE|Rotate|Jitter|Scale|Drop-G|Drop-L|Add-G|Add-L
> |:-:|:-:|:-:|:-:|:-:|:-:|:-:|:-:|:-:|:-:|:-:|
> |PointNext|Vanilla Training|0.873|0.921|0.995|1.079|0.803|0.807|0.942|0.944|0.875|
> ||Adv Training|0.870|0.901|0.991|1.027|0.803|0.833|0.929|0.912|0.809|
> ||DUP Defense|0.859|0.901|0.980|1.046|0.826|0.748|0.973|0.923|0.809|
> ||FAT (Ours)|0.875|**0.877**|0.998|0.916|0.791|0.786|0.867|0.938|0.840|
>
> The results demonstrate that FAT achieves consistent performance on the advanced network architecture PointNeXt, similar to observations on PointNet and more.
> Our FAT outperforms all other methods in terms of mCE.
> This indicates that FAT's performance is largely independent of the underlying model architecture, making it applicable to both traditional and modern networks. We will incorporate these additional experiments into the revision.
>
> ***Question 5: The paper lacks comparison with updated augmentation methods such as AdaptPoint, which could highlight the relative performance of FAT against newer augmentation techniques.***
>
> Thanks for the valuable suggestion. We further conduct experiments for comparison with AdaptPoint [2] on ScanObjectNN-C. AdaptPoint follows the official experimental settings [2]. The results are shown below.
>
> |ScanObjectNN-C|Method|OA|mCE|Rotate|Jitter|Scale|Drop-G|Drop-L|Add-G|Add-L
> |:-:|:-:|:-:|:-:|:-:|:-:|:-:|:-:|:-:|:-:|:-:|
> |PointNet|AdaptPoint|0.743|1.256|1.359|0.875|1.519|0.676|1.112|1.448|1.804|
> ||+ FAT|0.744|**1.196**|1.370|0.823|1.446|0.690|1.125|1.220|1.701|
> |PointNext|AdaptPoint|0.885|0.783|0.767|1.030|0.810|0.508|0.628|0.911|0.824|
> ||+ FAT|0.885|**0.761**|0.748|0.948|0.833|0.521|0.648|0.829|0.802|
>
> It is evident that incorporating FAT achieves a lower mCE, indicating its superiority. We will integrate these results into Table 2 and continue to include more comprehensive experiments as per your recommendations in the revised version.
>
> [1] Theoretically Principled Trade-off between Robustness and Accuracy, ICML 2019
>
> [2] Sample-adaptive Augmentation for Point Cloud Recognition Against Real-world Corruptions, ICCV 2023
>
> [3] Benchmarking Robustness of 3D Object Detection to Common Corruptions in Autonomous Driving, CVPR 2023
>
> [4] PointNeXt: Revisiting PointNet++ with Improved Training and Scaling Strategies, NeurIPS 2022

---

> > ### Comment · Reviewer_wa2k · 2024-08-08
> >
> > Thanks, updated my ratings! Good luck !

---

> > > ### Author Response · Authors · 2024-08-10
> > > **Thank you for increasing the score**
> > >
> > > Dear Reviewer wa2k,
> > >
> > > Thank you very much for increasing the score! We are glad to know that our response has addressed your concerns. We really appreciate your valuable feedback. We will further improve the paper in the final.
> > >
> > > Best regards, Authors

---

### Official Review · Reviewer_WzcS · 2024-07-10

**Soundness:** 3
**Presentation:** 3
**Contribution:** 3
**Rating:** 5
**Confidence:** 4

**Summary:**

This paper studies how to enhance the robustness of 3D point cloud recognition. The authors propose Frequency Adversarial Training (FAT) to improve the corruption robustness of 3D point cloud recognition models. FAT trains a model with adversarial examples that add perturbations to the frequency-domain representations of point clouds.

**Strengths:**

- The problem studied in this paper is important.

- The authors propose generating the adversarial sample in the frequency domain, which is interesting.

- This paper considers several baseline methods.

**Weaknesses:**

- This paper considers limited real-world applications. 3D point cloud recognition often fails due to LiDAR sensor inaccuracies and changes in the physical environment. However, this paper only utilizes one dataset, ModelNet40, where points are generated via 3D modeling techniques instead of being collected from real-world sensors, which is not convincing. If the paper aims to address real-world problems, it would be more convincing to include more real-world datasets, such as the KITTI dataset, where point clouds are collected from LiDAR sensors.

- The defense performance does not significantly outperform baseline methods. As shown in Table 1, FAT only outperforms baseline methods by a few points. Moreover, the proposed method appears to be effective only when combined with other data augmentation techniques. Could the authors provide more details and an explanation of the phenomenon of the ‘surprising results’ mentioned in the paper? For example, why does mixing methods result in much better defense?

- This paper does not analyze the relationship between the frequency and spatial domains in the context of the 3D point cloud. The two domains are mutually transformable, but the paper does not provide insights into their relationships or general observations.

- The paper only studies the defense effects against general 3D data corruptions without considering specific attack methods. It remains unclear if this approach can also defend against specific attack methods targeting 3D perception, such as attacks using adversarial points with specific shapes, locations, and rotation angles. The following are a few works about attacks against LiDAR object detection in autonomous driving scenarios [1-4] for reference.

[1] Yulong Cao, Chaowei Xiao, Dawei Yang, Jing Fang, Ruigang Yang, Mingyan Liu, and Bo Li. 2019. Adversarial objects against lidar-based autonomous driving systems. arXiv preprint arXiv:1907.05418 (2019).
[2] James Tu, Mengye Ren, Sivabalan Manivasagam, Ming Liang, Bin Yang, Richard Du, Frank Cheng, and Raquel Urtasun. 2020. Physically realizable adversarial examples for lidar object detection. In Proceedings of the IEEE/CVF conference on computer vision and pattern recognition.
[3] Yi Zhu, Chenglin Miao, Tianhang Zheng, Foad Hajiaghajani, Lu Su, and Chunming Qiao. 2021. Can we use arbitrary objects to attack lidar perception in autonomous driving?. In Proceedings of the 2021 ACM SIGSAC Conference on Computer and Communications Security.
[4] Shenchen Zhu, Yue Zhao, Kai Chen, Bo Wang, Hualong Ma, and Cheng’an Wei. 2024. AE-Morpher: Improve Physical Robustness of Adversarial Objects against LiDAR-based Detectors via Object Reconstruction. In Proceedings of the 33rd USENIX Security Symposium.

**Questions:**

Q1: As mentioned earlier in your paper, the corruption of 3D point clouds often occurs in real-world applications due to environmental noise and reflections. For instance, in autonomous driving scenarios, the LiDAR sensor may only capture partial points of the target vehicle because of occlusion and distance effects. However, this paper does not study those real-world data. I am curious whether your method would be effective on real-world datasets, such as the KITTI and Waymo driving datasets. Solely using ModelNet40 is not convincing.

Q2: Can the proposed approach maintain good performance when considering specific attack methods, such as those using adversarial points with specific shapes, locations, and rotation angles?

Q3: Modifications in the frequency domain can sometimes result in physically unrealizable transformations in the spatial domain. Will this limit the application of your work in real-world scenarios?

Q4: Can the authors provide more details on the reasons for using GFT rather than DCT or other frequency transformation methods?

Q5: What would be the outcome if medium frequency bands were modified? The proposed methods seem to only consider high and low frequencies.

**Limitations:**

Yes.

---

> ### Author Rebuttal · Authors · 2024-08-07
>
> Thank you for appreciating our new contributions as well as providing the valuable feedback. Below we address the detailed comments, and hope that you can find our response satisfactory.
>
> ***Question 1: The paper would be more convincing by using real-world data from LiDAR sensors, such as KITTI dataset.***
>
> Thanks for the valuable suggestion. We further conduct experiments on the KITTI and ScanObjectNN datasets, both collected by LiDAR sensors.
> ***Due to space constraints, detailed experimental results and analysis are presented in the Global Rebuttal.***
> Experiments on these two real-world datasets [1][2] further validate the superiority and practicality of our FAT.
>
> ***Question 2: Why does the proposed method appear to be significantly more effective when combined with other data augmentation techniques?***
>
> FAT operates in the frequency domain and considers the underlying structure of compactly represented 3D point clouds, while traditional data augmentation methods focus on spatial transformations within the 3D data itself.
> Therefore, attributed to the complementary and compatible information from both the spatial and frequency domains, mixing methods result in much better defense.
>
> ***Question 3: The paper does not analyze the relationship between the frequency and spatial domains in the context of the 3D point cloud.***
>
> Thanks for the suggestion.
> Similar to 2D images, the rough shape in point clouds’ spatial domain is represented by transformed low-frequency components while the fine details of objects are encoded in transformed high-frequency components.
> Furthermore, the frequency characteristics of point clouds represent higher level and global information than point-to-point relations in spatial domain. That is, the frequency representation encodes more abstract and essential contexts for recognizing the point cloud.
> This implies that in the frequency domain, point clouds are compactly represented, facilitating a better understanding of low-level distortions that are free of high-level semantics.
> We will incorporate this analysis in the revision.
>
> ***Question 4: The paper only studies the defense effects against general 3D data corruptions without considering specific attack methods.***
>
> Thanks for the suggestion.
> Actually, FAT can enhance the models’ robustness against adversarial attacks to some extent.
> The enhanced PointNet model on ModelNet40 achieves the adversarial accuracy of 31.2\% under PGD-20 at $\epsilon$ = 0.05, compared to 0\% for the standard trained PointNet.
> The enhanced PointPillars model on KITTI achieves a defense accuracy of 65.5\% with IoU greater than 0.7 against Tu's attack method [3], and 56.8\% against Zhu's attack method [4], compared to 42.5\% and 26.0\% for the standard trained PointPillars model.
> We will include more comprehensive experimental results and discussions on defending against adversarial attacks in the revision, including references and analyses of specific attack methods [3][4][5][6].
>
> ***Question 5: Modifications in the frequency domain can sometimes result in physically unrealizable transformations in the spatial domain. Will this limit the application of your work in real-world scenarios?***
>
> No, it does not limit applicability. As a training method, FAT only needs to consider transformations in the digital world, which is always feasible regardless of the scenario, unlike attack methods that must consider physical implementation.
>
> ***Question 6: Can the authors provide more details on the reasons for using GFT rather than DCT or other frequency transformation methods?***
>
> Images are typically transformed in the frequency domain with the 2D discrete Fourier transform (DFT) or discrete cosine transform (DCT). Different from images supported on regular grids, although 3D point clouds are highly structured, they reside on irregular domains without an ordering of points, hindering the deployment of traditional Fourier transforms.
> Specifically, an image represented as $\mathbb{R}^{n \times n}$ is regularly sampled on a grid, ensuring that pixels $(i, j)$ and $(i+1, j)$ are adjacent. In contrast, a point cloud represented as $\mathbb{R}^{n \times 3}$ lacks such ordering of points, where the Euclidean distance between the $i$-th and $(i+1)$-th points can be substantial. Traditional Fourier transforms cannot be directly applied to point clouds due to their unordered nature and loss of relative positional information.
> However, graphs provide a natural and accurate representation of irregular point clouds.
> Each point in a point cloud is treated as a vertex, connected to its $K$ nearest neighbors, with each point's coordinates serving as graph signals.
> Once a graph is constructed to represent the point cloud, the graph Fourier transform (GFT) can compactly transform it into the frequency domain by leveraging the edges of the graph to encode relative positional information.
>
> ***Question 7: What would be the outcome if medium frequency bands were modified?***
>
> Thanks for the suggestion. We further conduct more ablation studies among FAT, as well as FAT variants: FAT with only medium-frequency modified, FAT with only low-frequency modified, and FAT with only high-frequency modified.
> ***Due to space constraints, detailed experimental results and analysis are presented in the subsequent comments.***
> We will incorporate these additional experiments into the revision.

---

> > ### Author Response · Authors · 2024-08-07
> >
> > ***Question 7: What would be the outcome if medium frequency bands were modified?***
> >
> > Thanks for the suggestion. We further conduct more ablation studies among FAT, as well as FAT variants: FAT with only medium-frequency modified, FAT with only low-frequency modified, and FAT with only high-frequency modified. The results are shown below.
> >
> > |ModelNet-C|Method|OA|mCE|Rotate|Jitter|Scale|Drop-G|Drop-L|Add-G|Add-L
> > |:-:|:-:|:-:|:-:|:-:|:-:|:-:|:-:|:-:|:-:|:-:|
> > |PointNet|Vanilla Training|0.907|1.422|1.902|0.642|1.266|0.500|1.072|2.980|1.593|
> > ||FAT with only low-frequency modified|0.906|1.317|1.702|0.519|1.234|0.452|1.043|2.851|1.415|
> > ||FAT with only medium-frequency modified|0.905|1.342|1.729|0.439|1.468|0.500|1.144|2.757|1.354|
> > ||FAT with only high-frequency modified|0.890|1.306|1.614|0.373|1.734|0.504|1.193|2.627|1.098|
> > ||FAT|0.902|1.237|1.553|0.370|1.606|0.448|1.097|2.583|1.004|
> >
> > Compared with other methods, FAT with only high-frequency modified has a lower mCE for high-frequency corruptions such as “Jitter”, while FAT with only low-frequency modified has a lower mCE for low-frequency corruptions such as “scale”. As discussed in Sec. 3.3, this is because adversarial training on high/low frequencies reduces the high/low frequency sensitivity, thus improving robustness to high/low-frequency corruptions. The performance of FAT with only medium-frequency modified falls between FAT with only high-frequency modified and FAT with only low-frequency modified. Compared with these methods, FAT achieves the lowest mCE, showing the effectiveness of our algorithm. We will incorporate these additional experiments into the revision.
> >
> > [1] Sample-adaptive Augmentation for Point Cloud Recognition Against Real-world Corruptions, ICCV 2023
> >
> > [2] Benchmarking Robustness of 3D Object Detection to Common Corruptions in Autonomous Driving, CVPR 2023
> >
> > [3] Physically Realizable Adversarial Examples for Lidar Object Detection, CVPR 2020
> >
> > [4] Can We Use Arbitrary Objects to Attack Lidar Perception in Autonomous Driving? CCS 2021
> >
> > [5] Adversarial objects against lidar-based autonomous driving systems, arXiv
> >
> > [6] AE-Morpher: Improve Physical Robustness of Adversarial Objects against LiDAR-based Detectors via Object Reconstruction, USENIX 2024

---

> > > ### Comment · Reviewer_WzcS · 2024-08-10
> > > **Official Comment by Reviewer WzcS**
> > >
> > > Thanks for the rebuttal. I have updated the rating.

---

> > > > ### Author Response · Authors · 2024-08-14
> > > > **Thank you for increasing the score**
> > > >
> > > > Dear Reviewer WzcS,
> > > >
> > > > Thank you very much for increasing the score! We are glad to know that our response has addressed your concerns. We really appreciate you for spending considerable time on our paper. We will further improve the paper in the final.
> > > >
> > > > Best regards, Authors

---

### Official Review · Reviewer_kjym · 2024-07-11

**Soundness:** 4
**Presentation:** 4
**Contribution:** 3
**Rating:** 6
**Confidence:** 4

**Summary:**

This paper introduces a novel approach to improving the robustness of 3D point cloud recognition models by introducing Frequency Adversarial Training (FAT). By analyzing the frequency space of point clouds through the graph Fourier transform, the authors found that models are sensitive against different frequency bands of corruptions. To reduce the effect of both low and high frequency corruptions, the authors propose FAT, which introduces adversarial examples during training that were generated by perturbing the point cloud’s frequency space and taking the inverse graph Fourier transform. From the authors’ experiments, FAT significantly improved the robustness of point cloud recognition models and was able to achieve a new state-of-the-art performance.

**Strengths:**

The authors introduce an original analysis of point cloud recognition models by quantifying the sensitivity of these models against low-frequency disruptions, such as rotations, and high-frequency disruptions, such as jittering. Frequency adversarial training is also an innovative approach to improving the robustness of these models against these disruptions, utilizing the graph Fourier transform to adversarially train against low and high-frequency distortions within the frequency space. The writing, extensive experiments, and theoretical analysis provided by the authors effectively and clearly show the positive impact FAT has on the robustness of point cloud recognition models. The use of a point cloud’s frequency space and application of FAT have the capacity to have significant impacts on future research in point cloud models beyond 3D recognition models.

**Weaknesses:**

While the paper mentions the importance of robust models for real-world applications, it exclusively evaluates models trained with FAT on synthetic datasets like ModelNet-C and ModelNet40-C. Evaluating FAT using data from sensors such as LiDAR would further show FAT’s applicability under real-world conditions (such as KITTI dataset, if possible). Additionally, the paper does not discuss the complexity and efficiency of FAT, which are important factors for real-world applications.

**Questions:**

It will be great to see some results on real-world datasets such as KITTI.

---

> ### Author Rebuttal · Authors · 2024-08-07
>
> Thank you for appreciating our new contributions as well as providing the valuable feedback. Below we address the detailed comments, and hope that you can find our response satisfactory.
>
> ***Question 1: The paper would be more convincing by using real-world data from LiDAR sensors, such as KITTI dataset.***
>
> Thanks for the valuable suggestion. We further conduct experiments on the KITTI and ScanObjectNN datasets, both collected by LiDAR sensors.
> Experiments on these two real-world datasets further validate the superiority and practicality of our FAT.
> The experimental settings and evaluation metrics on ScanObjectNN-C [1] are consistent with those on ModelNet-C.
> The KITTI-C dataset [2] includes four major types of corruptions: Weather-level (e.g., Strong Sunlight), Sensor-level (e.g., Density Decrease), Motion-level (e.g., Moving Object), and Object-level (e.g., Local Gaussian Noise). Following [2], we use $AP_{cor}$ (corruption average precision) at moderate difficulty as the evaluation metric on KITTI-C, where higher values indicate better performance. We employ the representative 3D object detection model PointPillars. The results are shown below.
>
>
> |KITTI-C|Method|$AP_{clean}$|mean $AP_{cor}$|Weather-level $AP_{cor}$|Sensor-level $AP_{cor}$|Motion-level $AP_{cor}$|Object-level $AP_{cor}$|
> |:-:|:-:|:-:|:-:|:-:|:-:|:-:|:-:|
> |PointPillars|Vanilla Training|78.34|65.35|63.39|75.33|49.61|73.05|
> ||Adv Training|74.86|62.96|51.97|78.68|51.43|69.77|
> ||DUP Defense|72.02|63.98|58.19|78.05|48.21|71.46|
> ||FAT (Ours)|78.06|**67.02**|60.81|79.19|54.71|73.39|
>
>
> |ScanObjectNN-C|Method|OA|mCE|Rotate|Jitter|Scale|Drop-G|Drop-L|Add-G|Add-L
> |:-:|:-:|:-:|:-:|:-:|:-:|:-:|:-:|:-:|:-:|:-:|
> |DGCNN|Vanilla Training|0.858|1.000|1.000|1.000|1.000|1.000|1.000|1.000|1.000|
> ||Adv Training|0.843|1.062|1.146|0.778|1.097|0.873|0.960|1.185|1.396|
> ||DUP Defense|0.832|1.029|1.195|0.833|1.157|1.197|1.214|0.773|0.834|
> ||FAT (Ours)|0.856|**0.933**|0.968|0.815|0.959|0.852|0.950|0.959|1.026|
> |PointNet|Vanilla Training|0.739|1.354|1.610|0.884|1.427|0.786|1.264|1.487|2.022|
> ||Adv Training|0.725|1.334|1.532|0.844|1.403|0.873|1.333|1.474|1.881|
> ||DUP Defense|0.712|1.348|1.717|0.825|1.555|1.157|1.480|0.829|1.875|
> ||FAT (Ours)|0.734|**1.254**|1.393|0.796|1.465|0.844|1.264|1.259|1.759|
> |PointNext|Vanilla Training|0.873|0.921|0.995|1.079|0.803|0.807|0.942|0.944|0.875|
> ||Adv Training|0.870|0.901|0.991|1.027|0.803|0.833|0.929|0.912|0.809|
> ||DUP Defense|0.859|0.901|0.980|1.046|0.826|0.748|0.973|0.923|0.809|
> ||FAT (Ours)|0.875|**0.877**|0.998|0.916|0.791|0.786|0.867|0.938|0.840|
>
> It can be seen that our FAT generally leads to lower mCE (mean corruption error) on ScanObjectNN-C and higher $AP_{cor}$ (average precision) on KITTI-C. The experimental results on both KITTI and ScanObjectNN datasets further validate the generalizability and applicability of our FAT under real-world conditions. We will add the results in the revision.
>
> ***Question 2: The paper does not discuss the complexity and efficiency of FAT.***
>
> Thanks for the suggestion.
> The complexity of FAT implementation primarily involves generating high-frequency and low-frequency adversarial examples.
> Compared to standard adversarial training and other data augmentations, FAT incurs approximately $1.7 \sim 3.2$ times higher computational costs.
> Despite this limitation, such an increase in computational overhead is deemed acceptable for offline training scenarios.
> Moreover, our approach would not affect the efficiency of model inference, ensuring unhindered deployment of well-trained models in practical applications.
> We will add the discussion in the revision.
>
>
> [1] Sample-adaptive Augmentation for Point Cloud Recognition Against Real-world Corruptions, ICCV 2023
>
> [2] Benchmarking Robustness of 3D Object Detection to Common Corruptions in Autonomous Driving, CVPR 2023

---

> > ### Comment · Reviewer_kjym · 2024-08-09
> >
> > I am satisfied with the rebuttal and thus keep my rate unchanged.

---

> > > ### Author Response · Authors · 2024-08-10
> > > **Thank you for the appreciation of our contributions**
> > >
> > > Dear Reviewer kjym,
> > >
> > > We are pleased to know that you find our response satisfactory. We really appreciate your valuable comments. We will incorporate the additional experiments and improve the paper in the final version.
> > >
> > > Best regards, Authors

---

### Author Rebuttal · Authors · 2024-08-07

We deeply appreciate all the reviewers for their insightful and constructive reviews of our manuscript. Delightfully, we are glad that the reviewers found that:

- ***The presentation of our paper is polished and easy to understand.*** (Reviewers kjym, wa2k, ELyT)

- ***The problem studied in our paper is important*** for safety-critical applications such as autonomous driving and robotics. (Reviewers kjym, WzcS, wa2k)

- ***Our idea is novel, interesting, and well-motivated.*** (Reviewers kjym, WzcS, wa2k, ELyT)

- ***The performance of our apporach is well-justified, promising, and effective.*** (Reviewers kjym, wa2k, ELyT)

- ***The theoretical analysis of our paper offers valuable insights*** for analyzing model robustness. (Reviewers kjym, wa2k, ELyT)



Below we address a common concern raised by reviewers kjym, WzcS, and wa2k regarding the need for additional experiments on real-world LiDAR sensor datasets to enhance our paper's persuasiveness.

***Common Concern 1: The paper would be more convincing by additional experiments conducted on real-world LiDAR sensor datasets.***

Thanks for the valuable suggestion. We further conduct experiments on the KITTI and ScanObjectNN datasets, both collected by LiDAR sensors.
Experiments on these two real-world datasets further validate the superiority and practicality of our FAT.
The experimental settings and evaluation metrics on ScanObjectNN-C [1] are consistent with those on ModelNet-C.
The KITTI-C dataset [2] includes four major types of corruptions: Weather-level (e.g., Strong Sunlight), Sensor-level (e.g., Density Decrease), Motion-level (e.g., Moving Object), and Object-level (e.g., Local Gaussian Noise). Following [2], we use $AP_{cor}$ (corruption average precision) at moderate difficulty as the evaluation metric on KITTI-C, where higher values indicate better performance. We employ the representative 3D object detection model PointPillars. The results are shown below.

|KITTI-C|Method|$AP_{clean}$|mean $AP_{cor}$|Weather-level $AP_{cor}$|Sensor-level $AP_{cor}$|Motion-level $AP_{cor}$|Object-level $AP_{cor}$|
|:-:|:-:|:-:|:-:|:-:|:-:|:-:|:-:|
|PointPillars|Vanilla Training|78.34|65.35|63.39|75.33|49.61|73.05|
||Adv Training|74.86|62.96|51.97|78.68|51.43|69.77|
||DUP Defense|72.02|63.98|58.19|78.05|48.21|71.46|
||FAT (Ours)|78.06|**67.02**|60.81|79.19|54.71|73.39|

|ScanObjectNN-C|Method|OA|mCE|Rotate|Jitter|Scale|Drop-G|Drop-L|Add-G|Add-L
|:-:|:-:|:-:|:-:|:-:|:-:|:-:|:-:|:-:|:-:|:-:|
|DGCNN|Vanilla Training|0.858|1.000|1.000|1.000|1.000|1.000|1.000|1.000|1.000|
||Adv Training|0.843|1.062|1.146|0.778|1.097|0.873|0.960|1.185|1.396|
||DUP Defense|0.832|1.029|1.195|0.833|1.157|1.197|1.214|0.773|0.834|
||FAT (Ours)|0.856|**0.933**|0.968|0.815|0.959|0.852|0.950|0.959|1.026|
|PointNet|Vanilla Training|0.739|1.354|1.610|0.884|1.427|0.786|1.264|1.487|2.022|
||Adv Training|0.725|1.334|1.532|0.844|1.403|0.873|1.333|1.474|1.881|
||DUP Defense|0.712|1.348|1.717|0.825|1.555|1.157|1.480|0.829|1.875|
||FAT (Ours)|0.734|**1.254**|1.393|0.796|1.465|0.844|1.264|1.259|1.759|
|PointNext|Vanilla Training|0.873|0.921|0.995|1.079|0.803|0.807|0.942|0.944|0.875|
||Adv Training|0.870|0.901|0.991|1.027|0.803|0.833|0.929|0.912|0.809|
||DUP Defense|0.859|0.901|0.980|1.046|0.826|0.748|0.973|0.923|0.809|
||FAT (Ours)|0.875|**0.877**|0.998|0.916|0.791|0.786|0.867|0.938|0.840|

It can be seen that our FAT generally leads to lower mCE (mean corruption error) on ScanObjectNN-C and higher $AP_{cor}$ (average precision) on KITTI-C. The experimental results on both KITTI and ScanObjectNN datasets further validate the generalizability and applicability of our FAT under real-world conditions. We will add the results in the revision.

---

### Author Response · Authors · 2024-08-14
**Summary of Rebuttal**

Dear Reviewers, AC, and SAC:

We deeply thank the work done by AC and SAC such as distributing the paper to reviewers, guiding the reviewing process, and further supervising the discussion. We also sincerely appreciate the reviewers for taking the time to read our paper, providing constructive comments, and getting involved in our discussion. Without your elaborative help and support, our paper could not have been further polished.

Here we summarize our rebuttal to present a general perspective which could hopefully help grasp our contribution and modification quickly.

Through interactive discussion, all four reviewers have **found our response satisfactory**, with three of them **increasing their scores**.

The final scores given by the reviewers are as follows:
- **Soundness**: 4, 3, 3, 3
- **Presentation**: 4, 3, 3, 3
- **Contribution**: 3, 3, 3, 3
- **Rating**: 6, 6, 5, 5

Additionally, several consensuses have been achieved:
- **The presentation of our paper is polished, well-structured, and easy to understand.** (Reviewers kjym, wa2k, ELyT)
- **The problem studied in our paper is important and has significant impact** for safety-critical applications such as autonomous driving and robotics. (Reviewers kjym, WzcS, wa2k)
- **Our idea is novel, interesting, and well-motivated.** (Reviewers kjym, WzcS, wa2k, ELyT)
- **The performance of our approach is well-justified, promising, and effective.** (Reviewers kjym, wa2k, ELyT)
- **The theoretical analysis of our paper offers valuable insights** for analyzing model robustness. (Reviewers kjym, wa2k, ELyT)

Based on all the comments from the reviewers, we summarize the primary concerns and our responses as follows:

- ***Question 1: The paper would be more convincing by additional experiments conducted on real-world LiDAR sensor datasets.***

  In rebuttal, we further conduct experiments on the KITTI and ScanObjectNN datasets, both collected by LiDAR sensors. The results on these two real-world datasets further validate the superiority and practicality of our FAT.

- ***Question 2: The paper would be more comprehensive by additional experiments compared with updated network architectures/augmentation methods, and more ablation studies.***

  To address this, we further conduct experiments for PointNeXt (an advanced network architecture) and AdaptPoint (a recent augmentation method), along with additional ablation studies. These experiments offer a more thorough evaluation of our proposed FAT method.

- ***Question 3: The paper needs further explanation of the frequency and spatial domain aspects and the complexity and efficiency of FAT.***

  We have expanded our discussion on the relationship and impact of frequency and spatial domains, and provided a detailed analysis of the complexity and efficiency of FAT. We will incorporate the additional discussions and improve the paper in the final version.

In the past few weeks, we have tried our best to improve the quality of this paper and address each concern from all reviewers. We sincerely hope our effort can contribute to the community. Thanks again for your kind help and constructive opinions, we are truly grateful to have advice from you.

Sincerely,

Authors.

---

### Decision · Program_Chairs · 2024-09-25

**Decision:**

Accept (poster)

**Comment:**

This paper proposes a novel approach to improving the robustness of 3D point cloud recognition against corruptions by operating in the frequency domain. The method leverages the Graph Fourier Transform (GFT) to analyze and modify frequency components, enhancing corruption robustness. Extensive experiments demonstrate FAT's effectiveness across various network architectures and datasets, achieving state-of-the-art performance. The method is particularly relevant for safety-critical applications like autonomous driving.

The proposed method and the perspective to tackle 3D point cloud robustness is novel. The work offers strong theoretical grounding and empirical results, demonstrating the effectiveness on multiple architectures and datasets. Comprehensive experiments across various network architectures, including newer ones, and additional datasets such as KITTI and ScanObjectNN, are conducted. Although there were concerns regarding the limited real-world dataset usage (initially) and other ones such as complexity of implementation, impact on clean accuracy, and lack of comparison with specific attack methods, the value of the work outweighs its limitations. The authors have also effectively addressed concerns raised by reviewers, including adding real-world dataset experiments. These factors collectively justify an acceptance.